

# Locally-orthogonal unstructured grid-generation for general circulation modelling on the sphere[*]

Darren Engwirda[1,2]

[1]Department of Earth, Atmospheric and Planetary Sciences, Room 54-1517, Massachusetts Institute of Technology, 77 Massachusetts Avenue, Cambridge, MA 02139-4307
[2]NASA Goddard Institute for Space Studies, 2880 Broadway, New York, NY 10025 USA

*Correspondence to:* Darren Engwirda (engwirda@mit.edu)

**Abstract.** An algorithm for the generation of non-uniform, locally-orthogonal staggered unstructured grids on spheroidal geometries is described. This technique is designed to generate high-quality staggered Voronoi/Delaunay dual meshes appropriate for general circulation modelling on the sphere, including applications to atmospheric simulation, ocean-modelling and numerical weather prediction. Using a recently developed Frontal-Delaunay refinement technique, a method for the construc-

tion of unstructured spheroidal Delaunay triangulations is introduced. A locally-orthogonal polygonal grid, derived from the associated Voronoi diagram, is computed as the staggered dual. It is shown that use of the Delaunay-refinement technique allows for the generation of unstructured grids that satisfy a priori constraints on minimum mesh-quality. This initial staggered Voronoi/Delaunay tessellation is iteratively improved through hill-climbing optimisation techniques. It is shown that this approach typically produces grids with very high element quality and smooth grading characteristics, while imposing relatively

low computational expense. Initial results are presented for a selection of uniform and non-uniform spheroidal grids appropriate for high-resolution, multi-scale general circulation modelling. The use of user-defined mesh-spacing functions to generate smoothly graded, non-uniform grids for multi-resolution type studies is discussed in detail.

**Keywords.** Grid-generation; Frontal-Delaunay refinement; Voronoi tessellation; Grid-optimisation; Geophysical fluid dynamics; Ocean modelling; Atmospheric modelling; Numerical weather predication

# 1   Introduction

The development of atmospheric and oceanic general circulation models based on *unstructured* numerical discretisation schemes is an emerging area of research. This trend necessitates the development of unstructured grid-generation algorithms for the production of very high-resolution, *guaranteed-quality* unstructured triangular and polygonal meshes that satisfy non-uniform mesh-spacing distributions and embedded geometrical constraints. This study investigates the applicability of

a recently developed surface meshing algorithm (Engwirda and Ivers, 2016; Engwirda, 2016b) based on restricted Frontal-Delaunay refinement and hill-climbing type optimisation for this task.

---

[*]A short version of this paper appears in the proceedings of the 24th International Meshing Roundtable (Engwirda, 2015).




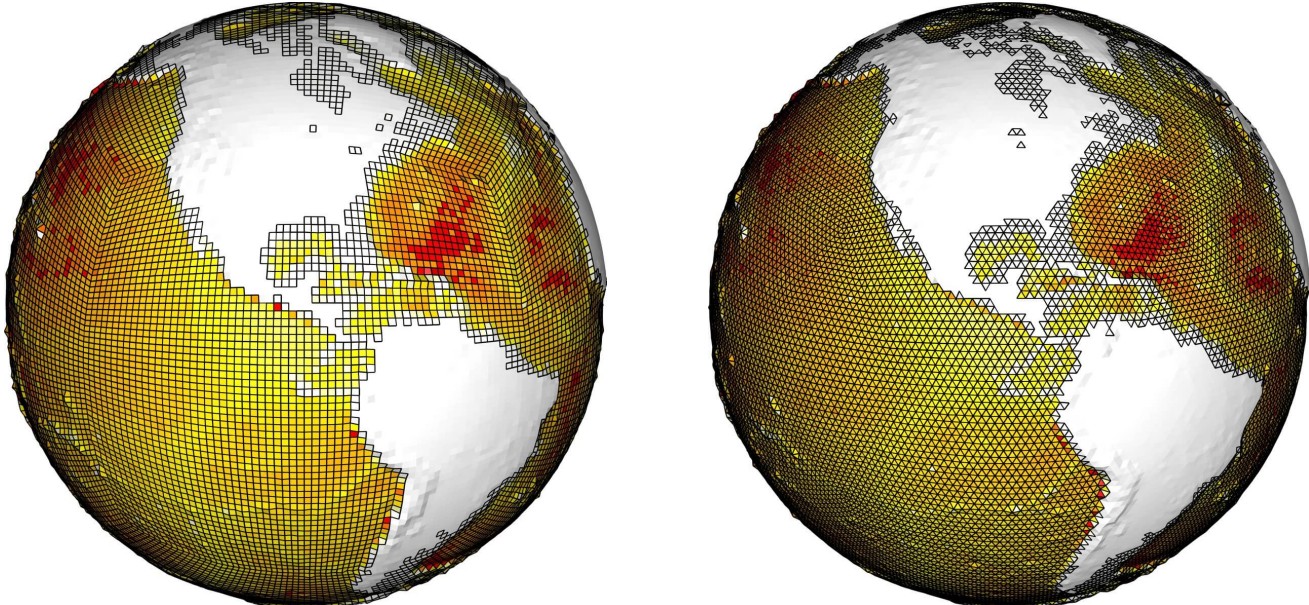

**Figure 1.** Conventional semi-structured meshing for the sphere, showing a regular cubed-sphere type grid (left), and a regular icosahedral class grid (right). Both grids were generated using equivalent target mean edge lengths, and are coloured according to mean topographic height at grid-cell centres. Topography is drawn using an exaggerated scale, with elevation from the reference geoid amplified by a factor of 20 in both cases.

## 1.1 Related Work

While simple structured grid types for the sphere can be obtained by assembling a uniform discretisation in spherical coordinates, the resulting *lat-lon* grid is inappropriate for numerical simulation, due to the presence of strong *grid-singularities* at the two poles. Such features manifest as local distortions in grid-quality, consisting of regions of highly distorted quadrilateral

grid-cells. These low-quality elements lead to a number of undesirable numerical effects — imposing restrictions on model time-step and stability, and compromising local spatial accuracy. A majority of current generation general circulation models, (i.e. Adcroft et al. (2004); Marshall et al. (1997); Putman and Lin (2007)), are instead based on a semi-structured quadrilateral discretisation known as the *cubed-sphere*. In this framework, the spherical surface is decomposed into a cube-like topology, with each of its six quadrilateral faces discretised as a structured curvilinear grid. In such an arrangement, the two strong grid-

singularities of the lat-lon configuration are replaced by eight weak discontinuities at the cube corners, leading to significant improvements in numerical performance. Putman and Lin (2007) present detailed discussions of techniques for the generation and optimisation of cube-sphere type grids. A regular *gnomonic-type* cubed-sphere grid is illustrated in Figure 1a.

In addition to the cubed-sphere type configuration, a second class of semi-structured spherical grid can be realised through *icosahedral-type* decompositions (Heikes and Randall, 1995; Randall et al., 2002). In such cases, the primary grid is defined as

a regular spherical triangulation, obtained through recursive bisection of the icosahedron. Additionally, a staggered polygonal



*dual* grid, consisting of hexagonal and pentagonal cells, is often used as a basis for finite-volume type numerical schemes. This geometrical duality is an example of the locally-orthogonal Delaunay/Voronoi type grid staggering that forms the basis of this paper. Icosahedral-type grids provide a near-perfect tessellation of the sphere — free of topological discontinuities and/or geometric irregularity. A regular icosahedral-class grid is illustrated in Figure 1b.

While both the cubed-sphere and icosahedral type grids provide an effective framework for uniform resolution global circulation models, the development of *multi-resolution* modelling environments has recently attracted considerable interest. A range of new general circulations models, including the Finite Element Sea Ice-Ocean Model (FESOM) (Wang et al., 2014), the Finite Volume Community Ocean Model (FVCOM) (Chen et al., 2003, 2007; Lai et al., 2010), the Stanford Unstructured Non-hydrostatic Terrain-following Adaptive Navier-Stokes Simulator (SUNTANS) (Fringer et al., 2006; Vitousek and Fringer, 

2014), and the Second-generation Louvain-la-Neuve Ice-ocean Model (SLIM) (Bernard et al., 2007; Comblen et al., 2009) are based on semi-structured triangular grids, with the horizontal directions discretised according to an unstructured spherical triangulation, and the vertical direction represented as a stack of locally structured layers. The Model for Predication Across Scales (MPAS) (Skamarock et al., 2012; Ringler et al., 2013, 2008) adopts a similar arrangement, except that a *locally-orthogonal* unstructured discretisation is adopted, consisting of both a Spherical Voronoi Tessellation (SVT) and its dual Delaunay trian-

gulation. The use of fully unstructured representations, based on general tetrahedral and/or polyhedral grids, are also under investigation in the Fluidity framework (Ford et al., 2004a, b; Pain et al., 2005; Piggott et al., 2008).

Existing approaches for unstructured grid-generation on spherical geometries have focused on a number of techniques, including: (i) the use of iterative, optimisation-type algorithms designed to construct Spherical Centroidal Voronoi Tessellations (SCVT's) (Jacobsen et al., 2013), and (ii) the adaptation of anisotropic two-dimensional meshing techniques (Lambrechts 

et al., 2008) that build grids in associated parametric spaces. The MPI-SCVT algorithm (Jacobsen et al., 2013) is a massively parallel implementation of iterative Lloyd-type smoothing (Du et al., 1999) for the construction of SVCT's for use in the MPAS framework. In this approach, a set of vertices are distributed over the spherical surface and iteratively *smoothed* until a high-quality Voronoi tessellation is obtained. Specifically, each iteration repositions vertices to the centroids of their associated Voronoi cells and updates the topology of the underlying spherical Delaunay triangulation. While such an approach 

typically leads to the generation of high-quality *centroidal* Voronoi tessellations on the sphere (SCVT's), the algorithm does not provide theoretical guarantees on minimum element quality, and often requires significant computational effort to achieve convergence. Additionally, current implementations of the MPI-SCVT algorithm do not provide a mechanism to constrain the grid to embedded features, such as coastal boundaries.

Lambrechts et al. (2008) present an unstructured spherical triangulation framework using the general-purpose grid-

generation package Gmsh (Geuzaine and Remacle, 2009). In this work, unstructured spherical triangulations are generated for the world ocean using a parametric meshing approach. Specifically, a triangulation of the spherical surface is generated by *mapping* the full domain (including coastlines) on to an associated two-dimensional parametric space via stereographic projection. Importantly, as a result of the projection, the grids constructed in parametric space must be highly anisotropic, such that a well-shaped, isotropic triangulation is induced on the sphere. A range of existing two-dimensional anisotropic meshing 

algorithms are investigated, including Delaunay-refinement, frontal, and adaptation-type approaches.





The current study explores the development of an algorithm for the generation of high-resolution, guaranteed-quality spheroidal Delaunay triangulations and associated Voronoi tessellations appropriate for a range of unstructured grid general circulation models. In this work, meshes are generated on the spheroidal surface directly, without local parameterisation or projection. Such an approach will be shown to exhibit significant flexibility — immune to issues of coordinate singularity and/or continental configuration. The applicability of this approach to grid-generation for imperfect spheres, including oblate spheroids and general ellipsoidal surfaces is explored. Adherence to user-defined mesh-spacing constraints is also discussed in detail.

## 2    A restricted Frontal-Delaunay refinement algorithm for spheroidal surfaces

The task is to generate very high-resolution, guaranteed-quality unstructured Delaunay triangulations for for planetary atmospheres and/or oceans. These grids will form a baseline for the hill-climbing mesh-optimisation methods presented in subsequent sections. In addition to bounds on minimum element quality, these meshes are also required to satisfy general non-uniform, user-defined mesh-spacing constraints. In this work, the applicability of a recently developed Frontal-Delaunay surface meshing algorithm (Engwirda and Ivers, 2016; Engwirda, 2016b) is investigated for this task.

An unstructured Delaunay triangulation of the reference spheroid associated with a general planetary geometry is sought. In a general form, this reference surface can be expressed as an axis-aligned triaxial ellipsoid

$$\sum_{i=1}^{3} \left( \frac{x_i}{r_i} \right)^2 = 1 \,, \tag{1}$$

where the $x_i$'s are the Cartesian coordinates in a locally aligned coordinate system, and the scalars $r_i > 0$ are its principal radii. Such a definition can be used to represent ellipsoidal surfaces in general position, based on the application of additional rigid-body translations and rotations.

### 2.1    Preliminaries

In this work, attention is restricted to the generation of locally-orthogonal staggered unstructured grids, consisting of Delaunay triangulations and their associated Voronoi duals. A full account of such structures is not presented here, instead, the reader is referred to the detailed theoretical exposition presented in, for example Cheng et al. (2013).

The Delaunay triangulation $\mathrm{Del}(X)$ associated with a set of points $X \in \mathbb{R}^d$ is characterised by the so-called *empty-circle* criterion — requiring that the set of circumscribing spheres $\mathrm{B}(\mathbf{c}_i, r_i)$ associated with each Delaunay triangle $\tau_i \in \mathrm{Del}(X)$ be *empty* of all points other than its own vertices. It is well known that for tessellations restricted to two-dimensional manifolds, the Delaunay triangulation leads to a maximisation of the minimum enclosed angle in the grid (Cheng et al., 2013). Such behaviour is clearly beneficial when seeking to construct high quality triangular meshes.

The Voronoi tessellation $\mathrm{Vor}(X)$ is the so-called *geometric-dual* associated with the Delaunay triangulation, consisting of a set of convex polygonal cells formed by connecting the centres of adjacent circumscribing balls — the so-called element *circumcentres* $\mathbf{c}_i$'s. The Voronoi tessellation represents a *closest-point map* for the points in $X$, with each Voronoi cell





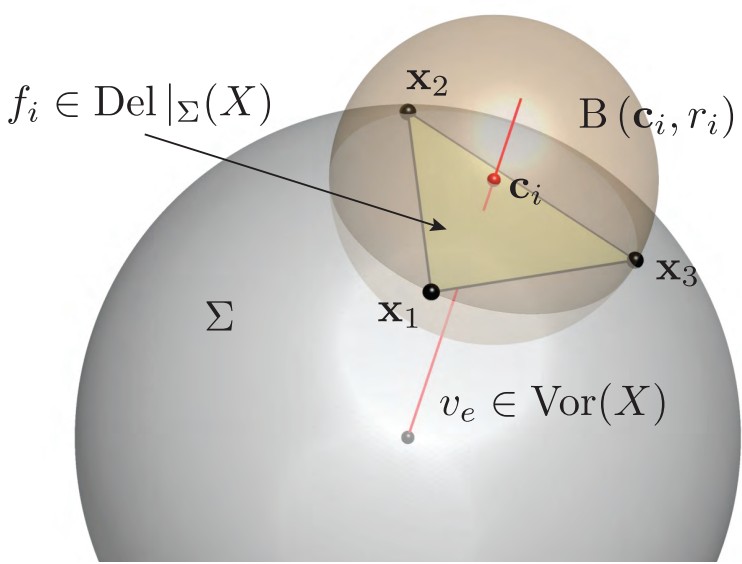

**Figure 2.** Illustration of the geometrical predicates used to identify restricted Delaunay surface triangules $f_i \in \mathrm{Del}|_\Sigma(X)$, showing details of the intersecting Voronoi edge $v_e \in \mathrm{Vor}(X)$ associated with the surface triangle $f_i \in \mathrm{Del}|_\Sigma(X)$ and its restricted surface-ball $\mathrm{B}(\mathbf{c}_i, r_i)$.

$v_c \in \mathrm{Vor}(X)$ defining the convex region adjacent to a given vertex $\mathbf{x}_i \in X$ for which $\mathbf{x}_i$ is the nearest point. Importantly, the Voronoi/Delaunay grid staggering, defines a *locally-orthogonal* arrangement, in which grid-cell edges in the Voronoi tessellation are orthogonal to adjacent edges in the underlying Delaunay triangulation. These orthogonality constraints make Voronoi/Delaunay type grids an attractive choice when seeking to construct staggered finite-volume type numerical discretisa-

tions. Such an approach is pursued by, for example, Ringler et al. in development of the MPAS modelling framework (Ringler et al., 2013, 2008).

## 2.2 Restricted Delaunay triangulation

In this study, grid-generation is carried out on the surface of the spheroidal geometry directly by making use of so-called *restricted* Delaunay mesh generation techniques. Specifically, given a reference spheroid $\Sigma$, grid-generation proceeds to dis-

cretise $\Sigma$ into a mesh of surface triangles. In the restricted Delaunay framework, a full-dimensional Delaunay triangulation $\mathrm{Del}(X)$ (i.e. a tetrahedral tessellation) is maintained, with surface triangles identified as the so-called *restricted* Delaunay surface-complex embedded in $\mathrm{Del}(X)$.

**Definition 1.** (restricted Delaunay tessellation) Let $\Sigma$ be a smooth surface embedded in $\mathbb{R}^3$. Let $\mathrm{Del}(X)$ be a full-dimensional Delaunay tetrahedralisation of a point-wise sample $X \subseteq \Sigma$ and $\mathrm{Vor}(X)$ be the associated Voronoi tessellation. The *restricted*

*Delaunay surface triangulation* $\mathrm{Del}|_\Sigma(X)$ is a sub-complex of $\mathrm{Del}(X)$ including any triangle $f_i \in \mathrm{Del}(X)$ associated with an *intersecting* Voronoi edge $\mathbf{v}_e \in \mathrm{Vor}(X)$ such that $\mathbf{v}_e \cap \Sigma \neq \emptyset$.





The development of restricted Delaunay techniques for general mesh-generation applications has been the subject of previous research, and a detailed discussion of such concepts is not presented here as a result. The reader is referred to the original work of Edelsbrunner and Shah (1997) or the detailed reviews presented in Cheng et al. (2013) for additional details and mathematical background.

5    In the context of this work, it is sufficient to note that the restricted Delaunay triangulation framework provides a convenient mechanism to identify the triangles that provide good approximations to the spheroidal surface $\Sigma$. An implementation of these ideas requires the definition of a single *geometrical-predicate*, designed to compute intersections between Voronoi edges and the underlying surface. In this study, this predicate is computed analytically, following standard spheroidal trigonometric manipulations, as detailed in Appendix A. See Figure 2 for detailed schematics.

## 2.3   Mesh-spacing functions

Local mesh density can be controlled via user-specified mesh-spacing functions $\bar{h}(\mathbf{x}) : \mathbb{R}^3 \to \mathbb{R}^+$ that define the *target* edge-length values over the domain to be meshed. In this work, mesh-spacing functions are specified as a discrete set of target values $\bar{h}_{i,j}$, defined on a simple background 'lat-lon' grid $\mathcal{G}$. The continuous mesh-spacing function $\bar{h}(\mathbf{x})$ is reconstructed using bilinear interpolation. As will be illustrated in subsequent sections, such an approach provides support for a wide range 15 of mesh-spacing definitions, including distributions derived from high-resolution topographic data (Amante and Eakins, 2009) or solution-adaptive metrics.

   In order to generate high-quality grids, it is necessary to ensure that the imposed mesh-spacing function is sufficiently smooth. Rather than requiring the user to accommodate such constraints, a Lipschitz smoothing process is adopted here. Following the work of Persson (2006), a *gradient-limited* mesh-spacing function $\bar{h}'(\mathbf{x})$ is constructed, by limiting the allowable 20 spatial fluctuation over each element in the background grid $\mathcal{G}$. In this study, a scalar smoothing parameter $g \in \mathbb{R}^+$ is used to limit variation, such that

$$\bar{h}'(\mathbf{x}_i) \le \bar{h}'(\mathbf{x}_j) + g \cdot \mathrm{dist}(\mathbf{x}_i, \mathbf{x}_j), \tag{2}$$

for all adjacent vertex pairs $\mathbf{x}_i$, $\mathbf{x}_j$ in $\mathcal{G}$. The gradient-limited mesh-spacing function $\bar{h}'(\mathbf{x})$ becomes more uniform as $g \to 0$. In this work, maximum gradient constraints are implemented following a *fast-marching* method, as described in Persson (2006).

## 25   2.4   Mesh-quality metrics

Before moving on to a detailed description of the grid-generation algorithm itself, a number of mesh-quality metrics are first introduced.

   **Definition 2.** (radius-edge ratio) Given a surface triangle $f_i \in \mathrm{Del}|_\Sigma(X)$, its *radius-edge* ratio, $\rho(f_i)$, is given by

$$\rho(f_i) = \frac{r_i}{\|\mathbf{e}_{\min}\|}, \tag{3}$$

30 where $r_i$ is the radius of the circumscribing ball associated with $f_i$ and $\|\mathbf{e}_{\min}\|$ is the length of its shortest edge.





The radius-edge ratio is a measure of element shape-quality. It achieves a minimum, $\rho(f_i) = 1/\sqrt{3}$ for equilateral triangles and increases toward $+\infty$ as elements tend toward degeneracy. The radius-edge ratio is directly related to the minimum plane-angle $\theta_{\min}$ between adjacent edges in the triangulation, such that $\rho(f_i) = \frac{1}{2}(\sin(\theta_{\min}))^{-1}$. Due to the summation of angles in a triangle, given a minimum angle $\theta_{\min}$ the largest angle $\theta_{\max}$ is also clearly bounded, such that $\theta_{\max} \leq \pi - 2\theta_{\min}$.

**Definition 3.** (area-length ratio) Given a surface triangle $f_i \in \mathrm{Del}|_\Sigma(X)$, its *area-length* ratio, $a(f_i)$, is given by

$$a(f_i) = \frac{4\sqrt{3}}{3} \frac{A_f}{\|\mathbf{e}_{\mathrm{rms}}\|^2},$$  (4)

where $A_f$ is the signed-area of $f_i$ and $\|\mathbf{e}_{\mathrm{rms}}\|$ is the root-mean-square edge length.

The area-length ratio is a robust, scalar measure of element shape-quality, and is typically normalised to achieve a score of $+1$ for ideal elements. The area-length ratio decreases with increasing distortion, achieving a score of $+0$ for degenerate

elements and $-1$ for fully inverted elements.

**Definition 4.** (relative edge-length) Given an edge in the surface tessellation $e_j \in \mathrm{Del}|_\Sigma(X)$, its *relative edge-length*, $h_r(e_j)$, is given by

$$h_r(e_j) = \frac{\|\mathbf{e}_j\|}{\bar{h}(\mathbf{x}_m)},$$  (5)

where $\|\mathbf{e}_j\|$ is the length of the $j$-th edge and $\bar{h}(\mathbf{x}_m)$ is the value of the mesh-spacing function sampled at the edge midpoint.

The relative-length distribution $h_r(e_j)$ is a measure of mesh-spacing conformance, expressing the ratio of actual-to-desired edge-length for all edge-segments in $\mathrm{Del}|_\Sigma(X)$. A value of $h_r(e_j) = 1$ indicates perfect mesh-spacing conformance.

## 2.5 Restricted Frontal-Delaunay refinement

In this study, a high-quality triangular surface mesh is generated on the spheroidal reference surface (1) using a *Frontal-Delaunay* variant of the conventional restricted Delaunay-refinement algorithm (Boissonnat and Oudot, 2003, 2005; Jamin

et al., 2015; Cheng et al., 2007, 2010). This technique is described by the author in detail in Engwirda and Ivers (2016); Engwirda (2016b) and differs from standard Delaunay-refinement approaches in terms of its strategies for the placement of new Steiner vertices. Specifically, the Frontal-Delaunay variant employs a generalisation of various *off-centre* type point-placement techniques (Rebay, 1993; Erten and Üngör, 2009), designed to position vertices so that element-quality and mesh-size constraints are satisfied in a *locally-optimal* fashion. Previous studies have shown that such an approach typically leads

to substantial improvements in mean element-quality and mesh smoothness. Additionally, it has been demonstrated that the Frontal-Delaunay method inherits much of the theoretical robustness of standard Delaunay-refinement techniques — offering guaranteed convergence, topological correctness, and minimum/maximum angle guarantees.

Given a user-defined mesh-spacing function $\bar{h}(\mathbf{x})$ and an upper-bound on the element *radius-edge* ratios $\bar{\rho} \geq 1$, the Frontal-Delaunay algorithm proceeds to sample the spheroidal surface $\Sigma$ by refining any surface triangle that violates either the mesh-

spacing or element-quality constraints. Refinement is accomplished by inserting a new Steiner vertex at the off-centre refinement point associated with a given element. Refinement continues until all constraints are satisfied. Upon termination, the resulting surface triangulation is guaranteed to contain nicely shaped elements, satisfying both the radius-edge constraints



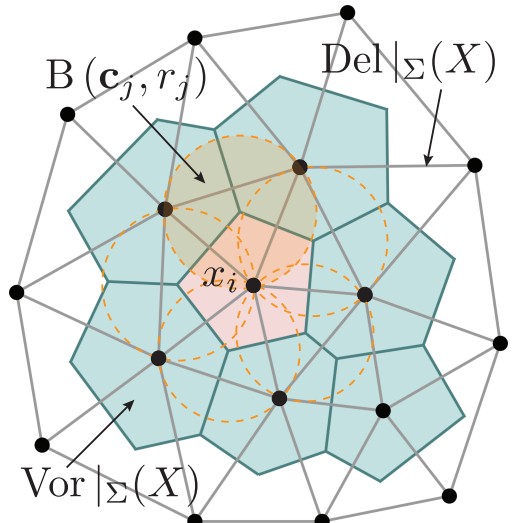
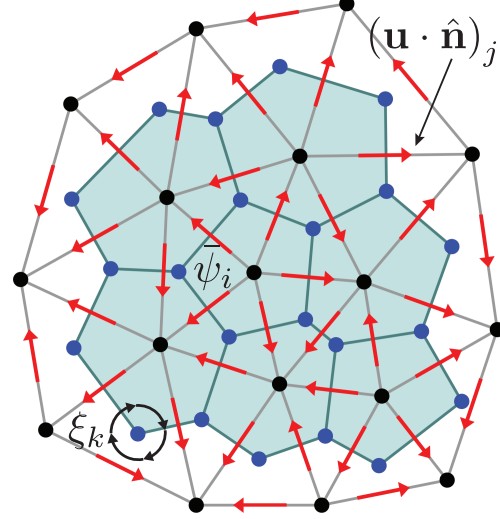

**Figure 3.** Construction of the staggered surface Voronoi control-volumes, illustrating: (left) locally-orthogonal Voronoi/Delaunay staggering, and (right) an associated unstructured C-grid type numerical discretisation scheme, as per the MPAS model. The formulation is a combination of conservative *cell-centred* tracer quantities $\bar{\psi}_i$, *edge-centred* normal velocity components $(\mathbf{u} \cdot \hat{\mathbf{n}})_j$, and auxiliary *vertex-centred* vorticity variables $\xi_k$.

$\rho(f_i) \leq \bar{\rho}$ and mesh-spacing bounds $h(f_i) \leq \bar{h}(\mathbf{x}_f)$ for all surface triangles $f_i \in \mathrm{Del}|_\Sigma(X)$ in the mesh. Setting $\bar{\rho} = 1$ guarantees that element angles are bounded, such that $30° \leq \theta_f \leq 120°$, ensuring that the grid does not contain any highly distorted elements. A full description of the Frontal-Delaunay refinement procedure — including a detailed discussion of its theoretical foundations — can be found in Engwirda and Ivers (2016); Engwirda (2016b).

As a *restricted* Delaunay-refinement approach, a full three-dimensional Delaunay tetrahedralisation $\mathrm{Del}(X)$ is incrementally maintained throughout the surface meshing phase, where $X \in \mathbb{R}^3$ is the set of vertices positioned on the surface of the spheroidal geometry. The set of restricted surface triangles $\mathrm{Del}|_\Sigma(X)$ that conform to the underlying spheroidal geometry are expressed as a subset of the tetrahedral faces $\mathrm{Del}|_\Sigma(X) \subseteq \mathrm{Del}(X)$. In an effort to minimise the expense associated with maintaining the full-dimensional topological tessellation, an additional *scaffolding* vertex $\mathbf{x}_s$ is initially inserted at the centre

of the spheroid. This has the effect of simplifying the resulting topological structure of the mesh, with the resulting tetrahedral elements forming a simple *wheel-like* configuration, in which they emanate radially outward from the central scaffolding vertex $\mathbf{x}_s$.

## 2.6  Construction of Voronoi cells

Typically, numerical formulations employed for large-scale atmospheric and/or oceanic general circulation modelling are based

on a *staggered* grid configuration, with quantities such as fluid pressure, geopotential, and density discretised using a primary control-volume, and the fluid velocity field and vorticity distribution represented on a second, spatially distinct grid-cell. In the





context of standard structured grid types, various staggered arrangements are described by the well-known Arakawa schemes (Arakawa and Lamb, 1977).

The development of general circulation models based on unstructured grid types is an emerging area of research, and, as a result, a variety of numerical formulations are currently under investigation. In this study, the development of *locally-orthogonal* grids appropriate for staggered unstructured numerical schemes are pursued, as these methods are thought to represent the most logical extension of the conventional structured Arakawa type techniques to the unstructured setting. Noting that the Frontal-Delaunay refinement algorithm described previously is guaranteed to construct triangulations that respect the Delaunay criterion, a natural staggered unstructured grid can be constructed based on the associated Voronoi diagram.

Consisting of a set of (convex) polygonal grid-cells centred on each vertex in the underlying triangulation, the surface Voronoi diagram $\mathrm{Vor}|_\Sigma(X)$ obeys a number of local orthogonality constraints. Specifically, grid-cell edges in the Voronoi tessellation are guaranteed to be perpendicular to their associated edges in the underlying Delaunay triangulation, passing through the Delaunay-edge midpoints. Additionally, in the case of perfectly *regular* and *centroidal* tessellations, the Delaunay-edges are guaranteed to pass through the midpoints of their associated Voronoi duals. Voronoi grid-cells are formed as the convex-hull of the incident element surface-ball centres $\mathrm{B}(\mathbf{c}_i, r_i)$ associated with the set of surface triangles adjacent to each vertex. Example Voronoi/Delaunay type grid staggering is illustrated in Figure 3.

While detailed comparisons of particular numerical discretisation schemes lie outside the scope of the current study, brief comments regarding the benefits of locally-orthogonal grid-staggering arrangements are made. Pursuing an unstructured variant of the widely-used Arakawa C-grid, the placement of fluid pressure, geopotential and density degrees-of-freedom within the primary Voronoi control-volumes, and orthogonal velocity vectors on Delaunay-edges achieves a similar configuration. Such an arrangement facilitates construction of a standard conservative finite-volume type scheme for the transport of fluid properties and a *mimetic* class (Lipnikov et al., 2014; Bochev and Hyman, 2006) finite-difference formulation for velocity components. Additionally, exploiting alignment with Delaunay-edges, a conservative evaluation of the fluid vorticity can be made on the staggered Delaunay triangles. This type of *unstructured* C-grid staggering is employed in the Model for Prediction Across Scales (MPAS), for both atmospheric and oceanic modelling (Skamarock et al., 2012; Ringler et al., 2013, 2010). See Figure 3 for additional details.

## 3   Hill-climbing mesh optimisation

While the staggered Voronoi/Delaunay grids generated using the Frontal-Delaunay refinement algorithm described in Section 2 are guaranteed to be of very high-quality, these tessellations can often be further improved through subsequent *mesh-optimisation* operations. Such a procedure is realised as a coupled geometrical and topological optimisation task — seeking to reposition vertices and update grid topology to maximise a given element-wise mesh-quality metric $\mathcal{Q}_f(\mathbf{x})$. In this study, a *hill-climbing* type optimisation strategy is pursued, in which a locally-optimal solution is sought based on an initial grid configuration.



In the present work, the grid is optimised according to the *area-length* quality metric (2.4), a robust scalar measure of mesh-quality that achieves a score of $+1$ for 'perfect' elements — decreasing toward zero with increasing levels of element distortion. Optimisation predicates are implemented in a so-called *hill-climbing* fashion (Freitag and Ollivier-Gooch, 1997; Klingner and Shewchuk, 2008), with modifications to the grid accepted only if the local mesh-quality metrics are sufficiently improved.

Specifically, a *worst-first* strategy is adopted, in which each given optimisation predicate is required to improve the *worst-case* quality associated with elements in the local subset being acted upon. Such a philosophy ensures that global mesh-quality is increased monotonically as optimisation proceeds. Note that such behaviour is designed to maximise the minimum element quality metric in the grid, rather than improving a mean measure. This represents an important distinction when compared to other iterative mesh-optimisation algorithms, such as Centroidal Voronoi Tessellation (CVT) type schemes (Jacobsen et al.,

2013), in which all nodes are typically adjusted simultaneously until a global convergence criterion is satisfied.

### 3.1 'Spring'-based mesh smoothing

Considering firstly the geometric optimality of the grid, a *mesh-smoothing* procedure is undertaken — seeking to reposition the nodes of the grid to improve element quality and mesh-spacing conformance. Following the work of Persson and Strang (2004), a *spring-based* approach is pursued, in which edges in the Delaunay triangulation are treated as *elastic-rods* with a

prescribed natural length. Nodes are iteratively repositioned until a local equilibrium configuration is reached. In the original work of Persson, nodal positions are adjusted via a local time-stepping loop, with all nodes updated concurrently under the action of explicit spring forces. In the current study, a non-iterative variant is employed, in which each node is repositioned one-by-one, such that constraints in each local neighbourhood are satisfied directly. Specifically, a given node $\mathbf{x}_i$ is repositioned as a weighted sum of contributions from incident edges

$$\mathbf{x}_i^{n+1} = \frac{\sum w_k(\mathbf{x}_i^n + \Delta_k \mathbf{v}_k)}{\sum w_k}, \qquad \text{where:} \quad \mathbf{v}_k = \mathbf{x}_i^n - \mathbf{x}_j^n, \quad \Delta_k = \frac{\bar{h}(\mathbf{x}_k^n) - l_k}{l_k}. \qquad (6)$$

Here, $\mathbf{x}_i^n$, $\mathbf{x}_j^n$ are the current positions of the two nodes associated with the $k$-th edge, $l_k$ is the edge length and $\bar{h}(\mathbf{x}_k)$ is the value of the mesh-spacing function evaluated, at the edge midpoint. $\Delta_k$ is the relative spring *extension* required to achieve equilibrium in the $k$-th edge. The scalars $w_k \in \mathbb{R}^+$ are edge weights. Setting $w_k = 1$ results in an unweighted scheme, consisting of simple *linear* springs. In this study, the use of nonlinear weights, defined by setting $w_k = \Delta_k^2$, was found to offer superior performance.

Noting that application of the spring-based operator (6) may move nodes away from the underlying spheroidal surface $\Sigma$, an additional *projection* operator is introduced to ensure that the grid conforms to the surface geometry exactly. Following the application of each spring-based adjustment (6), nodes are moved back onto the geometry via a closest-point projection.

Consistent with the hill-climbing paradigm described previously, each nodal adjustment (6) is required to be *validated* before being committed to the updated grid configuration. Specifically, nodal adjustments are accepted only if there is sufficient

improvement in the mesh-quality metrics associated with the set of adjacent elements. A sorted comparison of quality metrics before and after nodal repositioning is performed, with nodal adjustments successful if grid-quality is improved in a *worst-first* manner. This *lexicographical* quality comparison is consistent with the methodology employed in Klingner and Shewchuk (2008).





## 3.2 Gradient-based mesh smoothing

While the spring-based mesh-smoothing operator described previously is effective in adjusting a grid to satisfy mesh-spacing constraints, and tends to improve mesh-quality on average, it is not guaranteed to improve worst-case element quality metrics in all cases. As such, an additional *steepest-ascent* type optimisation strategy is pursued (Freitag and Ollivier-Gooch, 1997), in

which nodal positions are adjusted using the local gradients of incident element quality functions. Specifically, a given node $\mathbf{x}_i$ is repositioned along a local *search-vector* chosen to improve the quality of the worst incident element

$$\mathbf{x}_i^{n+1} = \mathbf{x}_i^n + \Delta_f \hat{\mathbf{v}}_f, \qquad \text{where:} \quad \mathbf{v}_f = \frac{\partial}{\partial \mathbf{x}}(\mathcal{Q}_f(\mathbf{x}_i)), \quad f = \arg\min_j(\mathcal{Q}_j(\mathbf{x}_i)). \tag{7}$$

Here, the index $j$ is taken as a loop over the Delaunay triangles $f_j \in \mathrm{Del}|_\Sigma(X)$ incident to the node $\mathbf{x}_i$. The scalar step-length $\Delta_f \in \mathbb{R}^+$ is computed via a line-search along the gradient ascent vector $\hat{\mathbf{v}}_f$, and in this study, is taken as the first value found

that leads to a net improvement in the worst-case incident quality metric $\mathcal{Q}_f(\mathbf{x}_i)$. A simple bisection-type strategy is used in the present work, iteratively testing $\Delta_f^p = \left(\frac{1}{2}\right)^p \alpha$ until a successful nodal adjustment is found. Here $p$ denotes the local line-search iteration. The scalar length $\alpha \in \mathbb{R}^+$ is determined as a solution to the first-order Taylor expansion

$$\mathcal{Q}_f(\mathbf{x}_i) + \alpha \, \hat{\mathbf{v}}_f \cdot \frac{\partial}{\partial \mathbf{x}}(\mathcal{Q}_f(\mathbf{x}_i)) \leq \tilde{\mathcal{Q}}_j(\mathbf{x}_i). \tag{8}$$

The index $j$ is again taken as a loop over the Delaunay triangles $f_j \in \mathrm{Del}|_\Sigma(X)$ incident to the central node $\mathbf{x}_i$, with the quantity

$\tilde{\mathcal{Q}}_j(\mathbf{x}_i)$ representing the *second-lowest* adjacent grid-quality score. This selection strategy (Freitag and Ollivier-Gooch, 1997) is designed to compute an initial displacement $\alpha$ that will improve the worst element in the adjacent set until its quality is equal that of its next best neighbour. Noting that such an expansion is only first-order accurate, the step-length is iteratively decreased using bisection. In this study, a limited line-search is employed, testing iterations $p = \{0, 1, \ldots, 5\}$ until a successful step is found. Consistent with the spring-based procedure described previously, a geometry-projection operator is implicitly

incorporated within each update (7), ensuring that nodes remain constrained to the spheroidal surface.

## 3.3 Topological 'flips'

In addition to purely geometrical operations, general grid optimisation also requires that adjustments be made to the underlying mesh topology, such that the surface triangulation remains a valid Delaunay structure. While it is possible to simply re-compute the full restricted Delaunay topology after each nodal position adjustment, such an approach carries significant computational

costs, especially when considering that a majority of nodal position updates involve small perturbations. In this work, an alternative strategy is pursued, based on local element-wise transformations, known as topological *flips*.

For any given pair of adjacent surface triangles $f_i, f_j \in \mathrm{Del}|_\Sigma(X)$, a local re-triangulation can be achieved by *flipping* local connectivity about the shared edge $x_i, x_j$ to instead form a new edge between the opposing vertices $x_a, x_b$. Such an operation results in the deletion of the existing triangles $f_i, f_j$ and the creation of a new pair $f_i', f_j'$. This operation is illustrated in

Figure 4a. In the present study, the iterative application of such *edge-flipping* operations is used to adjust the topology of the surface triangulation, such that it remains Delaunay. Specifically, given a general, possibly non-Delaunay, surface triangulation



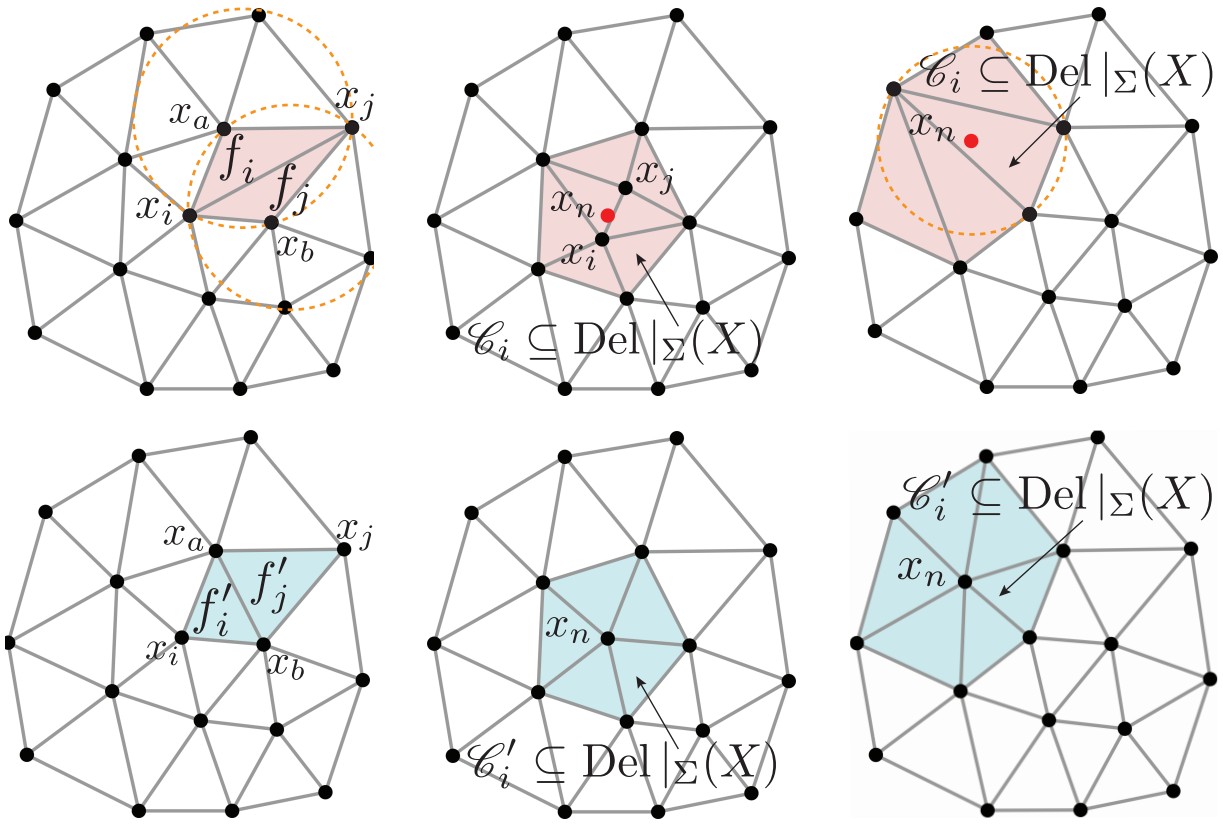

**Figure 4.** Topological operations for grid optimsiation, showing (left) an edge-flip, (middle) an edge-contraction, and (c) an edge-refinement operation. Grid configurations before and after each flip are shown in the upper and lower panels, respectively.

$\mathrm{Tri}|_\Sigma(X)$, a cascade of edge-flips are used to achieve a valid restricted Delaunay surface tessellation $\mathrm{Del}|_\Sigma(X)$. For each adjacent triangle pair $f_i, f_j \in \mathrm{Tri}|_\Sigma(X)$ an edge-flip is undertaken if a local violation of the Delaunay criterion is detected. New elements created by successful edge-flips are iteratively re-examined until no further modifications are necessary. This approach follows the standard flip-based algorithms described in, for instance Lawson (1977); Cheng et al. (2013).

5    Given a triangle $f_i \in \mathrm{Tri}|_\Sigma(X)$, the local Delaunay criterion is violated if there exists a node $x_q \notin f_i$ *interior* to the circumscribing ball associated with the triangle $f_i$. In this work, violations to the Delaunay criterion are detected by considering the *restricted* circumballs $\mathrm{B}(\mathbf{c}_i, r_i)$ associated with each triangle $f_i \in \mathrm{Tri}|_\Sigma(X)$, where the ball-centre $\mathbf{c}_i$ is a projection of the *planar* element circumcentre onto the spheroidal surface $\Sigma$. Such constructions account for the curvature of the surface. Given an adjacent triangle pair $f_i, f_j \in \mathrm{Tri}|_\Sigma(X)$ an edge-flip is undertaken if either opposing vertex $x_a, x_b$ lies within the

10    circumball associated with the adjacent triangle. To prevent issues associated with exact floating-point comparisons, a small relative tolerance is incorporated. Specifically, nodes are required to penetrate the opposing circumball by a distance $\epsilon$ before an edge-flip is undertaken, with $\epsilon = \frac{1}{2}(r_i + r_j)\bar{\epsilon}$ and $\bar{\epsilon} = 1 \times 10^{-10}$ in the current double-precision implementation.



### 3.4 Edge contraction

In some cases, grid-quality and mesh-spacing conformance can be improved through the use of so-called *edge-contraction* operations, whereby nodes are removed from the grid by collapsing certain edges. Given an edge $e_k \in \mathrm{Tri}|_\Sigma(X)$, a re-triangulation of the local *cavity* $\mathscr{C}_i \subseteq \mathrm{Tri}|_\Sigma(X)$, formed by the set of triangles incident to the nodes $x_i, x_j \in e_k$, can be achieved

by *merging* the nodes $x_i, x_j$ at some midpoint along the edge $e_k$. In addition to collapsing the edge $e_k$, edge-contraction also removes the two surface triangles $f_i, f_j \in \mathrm{Tri}|_\Sigma(X)$ adjacent to $e_k$, resulting in a new re-triangulation of the local cavity $\mathscr{C}_i' \subseteq \mathrm{Tri}|_\Sigma(X)$. See Figure 4b for illustration. In the present work, nodes are merged to a mean position $\mathbf{x}_n$ — taken as an average of the adjacent element circumcentres, such that $\mathbf{x}_n = \frac{1}{|\mathscr{C}_i|} \sum \mathbf{c}_j$, where the $\mathbf{c}_j$'s are centres of the circumballs associated with the adjacent surface triangles $f_j \in \mathscr{C}_i$. The mean position $\mathbf{x}_n$ is projected onto the spheroidal surface $\Sigma$. While such

an approach is slightly more computationally intensive than use of the simple edge-midpoint, the local circumcentre-based strategy proved to be substantially more effective in practice. Consistent with the hill-climbing philosophy pursued throughout this study, edge-contraction operations are only successful if there is sufficient improvement in the mesh-quality metrics associated with the set of adjacent elements. As per previous discussions, edge-contraction is undertaken based on a lexicographical comparison of the grid-quality vectors associated with the initial and final grid states $\mathscr{C}_i$ and $\mathscr{C}_i'$, respectively.

### 3.5 Edge refinement

Fulfilling the opposite role to edge-contraction, so-called *edge-refinement* operations seek to improve grid-quality and mesh-spacing conformance through the addition of new nodes and elements. In the present study, a simplified refinement operation is utilised, in which a given edge $e_k \in \mathrm{Tri}|_\Sigma(X)$ is refined by placing a new node $x_n$ at the centre of the restricted circumball $\mathrm{B}(\mathbf{c}_i, r_i)$ associated with the lower quality adjacent triangle $f_i \in \mathrm{Tri}|_\Sigma(X)$. Insertion of the new node $x_n$ induces the re-

triangulation of a local cavity $\mathscr{C}_i \in \mathrm{Tri}|_\Sigma(X)$ — constructed by expanding about $x_n$ in a local greedy fashion. Starting from the initial cavity $\mathscr{C}_i = \{f_i, f_j\}$ adjacent to the edge $e_k$, additional elements are added in a *breadth-first* manner, with a new, unvisited neighbouring element $f_k$ added to the cavity $\mathscr{C}_i$ if doing so will improve the worst-case element quality metric. The final cavity $\mathscr{C}_i$ is therefore a locally-optimal configuration. In practice, the iterative deepening of $\mathscr{C}_i$ typically convergences in one or two iterations. See Figure 4c for illustration. As per the edge-contraction and node-smoothing operations described

previously, edge-refinement is implemented according to a hill-climbing type philosophy, with operations successful only if there is sufficient improvement in local grid-quality. Consistent with previous discussions, a lexicographical comparison of the grid-quality metrics associated with elements in the initial and final states $\mathscr{C}_i$ and $\mathscr{C}_i'$ is used to determine success.

### 3.6 Optimisation schedule

The full grid optimisation procedure is realised as a combination of the various geometrical and topological operations de-

scribed previously, organised into a particular iterative optimisation *schedule*. Each outer iteration consists of a fixed set of operations: four node-smoothing passes, a single pass of edge refinement/contraction operations, and, finally, iterative edge-flipping to restore the Delaunay criterion. In this study, sixteen outer iterations are employed. Each node-smoothing pass is a



composite operation, with the spring-based technique used to adjust nodes adjacent to high-quality elements, and the gradient-ascent method used otherwise. Specifically, spring-based smoothing is used to adjust nodes adjacent to elements with a minimum quality score of $\mathcal{Q}_f \geq 0.9375$. Such thresholding ensures that the expensive gradient-ascent type iteration is reserved for the worst elements in the grid. The optimisation schedule employed here is not based on any rigorous theoretical derivation,

but is simply a set of heuristic choices that have proven to be effective in practice. The application of multiple node-smoothing passes within an outer iteration containing subsequent topological, contraction and refinement operations is consistent with the methodologies employed in, for instance Freitag and Ollivier-Gooch (1997); Klingner and Shewchuk (2008).

## 4    Results & Discussions

The performance of the Frontal-Delaunay refinement and hill-climbing optimisation algorithms presented in Sections 2 and

3 was investigated experimentally, with the methods used to mesh a series of benchmark problems. The algorithms were implemented in C++ and compiled as a 64-bit executable. The full algorithm has been implemented as a specialised variant of the general-purpose JIGSAW meshing package, denoted JIGSAW-GEO, and is currently available online (Engwirda, 2016a) or by request from the author. All tests were completed on a Linux platform using a single core of an Intel i7 processor. Visualisation and post-processing was completed using MATLAB.

### 4.1    Preliminaries

The JIGSAW-GEO algorithm was used to mesh a set of benchmark problems, suitable for various atmospheric and oceanic general circulation problems. The UNIFORM-SPHERE test-case describes a uniform resolution meshing problem on the sphere, suitable for uniformly resolved atmospheric and/or oceanic studies. The REGIONAL-ATLANTIC test-case describes a simple, regionally-refined grid for global ocean modelling, incorporating a high-resolution, eddy-permitting representation

of the North Atlantic ocean basin. Lastly, the SOUTHERN-OCEAN test-case describes a multi-resolution, regionally-refined grid for global ocean simulation, with a very high-resolution representation of the Southern Ocean and Antarctic regions. The mesh-spacing function for this problem was designed using a combination of topographic gradients and regional-refinement. The Voronoi/Delaunay grids for these test-cases are shown in Figures 5, 7, 9 and 11 with associated grid-quality statistics presented in Figures 6, 8 and 10.

In all test cases, limiting radius-edge ratios were specified, such that $\bar{\rho}_f = 1.05$. These constraints ensure that the minimum enclosed angle in any triangle is $\theta_{\min} \geq 28.4°$. For all test problems, detailed statistics on element quality are presented, including histograms of element *area-length ratios* $a_f$, *element-angles* $\theta_f$, and *relative-edge-length* $h_r$. The element area-length ratios are robust measures of element quality, where high-quality elements attain scores that approach unity. The relative edge-length metric is defined to be the ratio of the measured edge-length $\|\mathbf{e}\|$ to the target value $\bar{h}(\mathbf{x}_e)$, where $\mathbf{x}_e$ is the edge midpoint.

Relative edge-lengths close to unity indicate conformance to the imposed mesh-spacing function. High-quality surface triangles contain angles approaching $60°$. Histograms further highlight the minimum, maximum and mean values of the relevant distributions as appropriate.





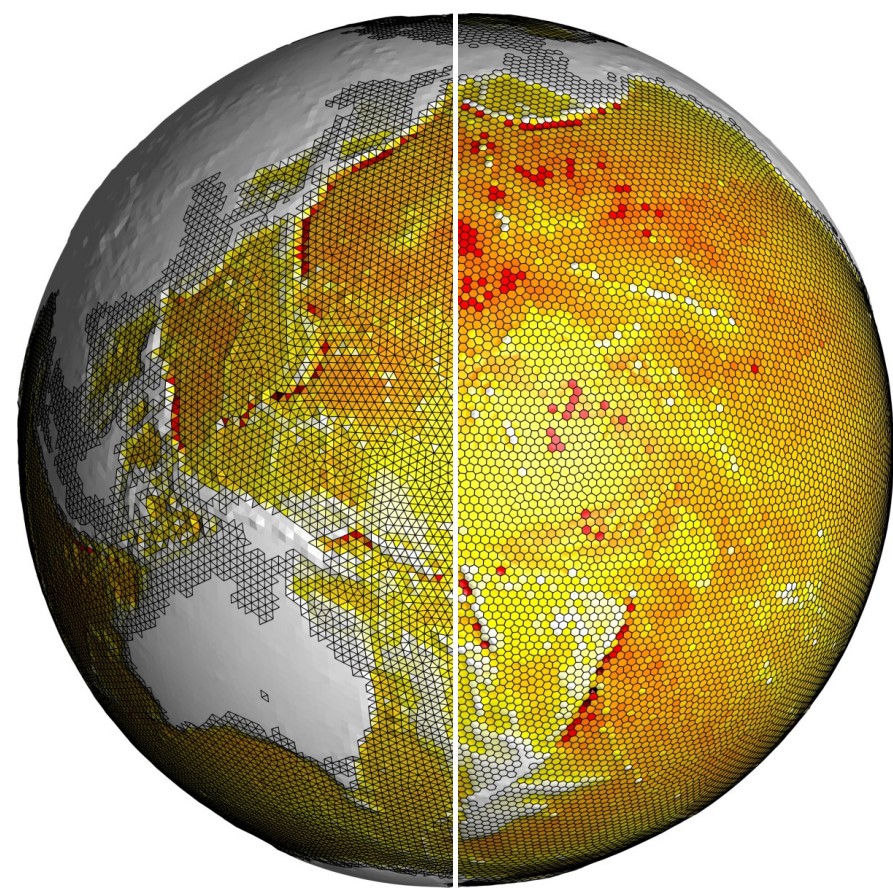

**Figure 5.** A uniform resolution global grid, showing (left) the underlying spheroidal Delaunay triangulation, and (right) the associated staggered Voronoi dual. $150\,\mathrm{km}$ grid-spacing was specified globally. Topography is drawn using an exaggerated scale, with elevation from the reference geoid amplified by a factor of 10 in all cases.

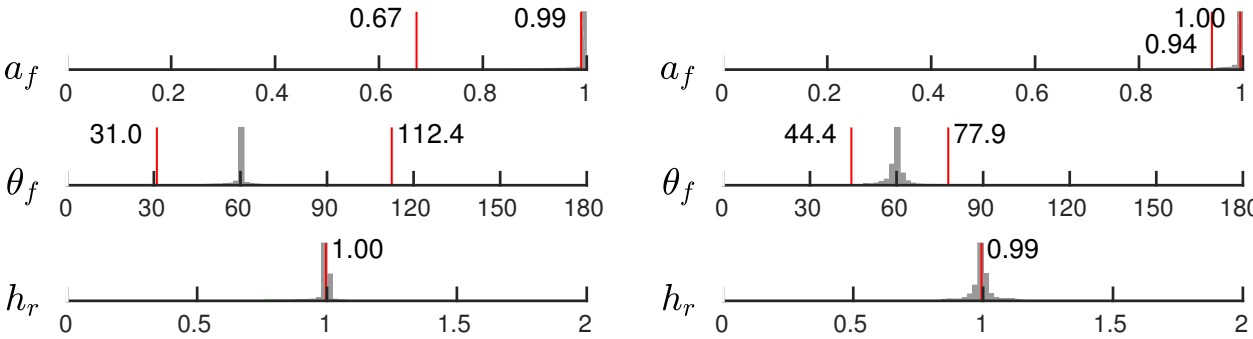

**Figure 6.** Mesh-quality metrics associated with the uniform resolution global grid, before (left) and after (right) the application of hill-climbing mesh optimisation. Normalised histograms of element area-length ratio $a_f$, enclosed-angle $\theta_f$ and relative-length $h_r$ are illustrated, with minimum, maximum and mean values annotated.



## 4.2 Uniform global grid

The performance of the JIGSAW-GEO algorithm was first assessed using the UNIFORM-SPHERE test-case, seeking to build a uniformly resolved, staggered Voronoi/Delaunay-type dual grid for general circulation modelling. Spatially uniform mesh-size constraints were enforced, setting $\bar{h}(\mathbf{x}) = 150\,\mathrm{km}$ over the full sphere. The resulting grid is shown in Figure 5 and contains
83,072 Delaunay triangles and 41,538 Voronoi cells. Grid-generation time was approximately 12 seconds, including both the initial restricted Frontal-Delaunay refinement and subsequent hill-climbing type optimisation. Grid-quality metrics are presented in Figure 6, showing distributions before and after the application of the grid-optimisation procedure.

Overall, the high quality of the Voronoi/Delaunay grids presented in Figure 5 illustrates the effectiveness of the JIGSAW-GEO algorithm. Based on visual inspection, it is clear that the grids achieve very high levels of geometric quality — being
absent of distorted grid-cell configurations and/or areas of over- or under-refinement. Focusing on the distribution of triangle shape-quality explicitly, it is noted that very high levels of mesh regularity are achieved, with the vast majority of element area-length scores tightly clustered about $a_f = 1$. Similarly, the distribution of element angles shows strong convergence around $\theta_f = 60°$, revealing most triangles to be near equilateral. Finally, analysis of the relative-length distributions show that grids tightly conform to the imposed mesh-spacing constraints, with a tight clustering of $h_r$ about 1.
The effect of the grid-optimisation procedure can be assessed by comparing the mesh-quality statistics presented in Figure 6. The application of mesh-optimisation is seen to be most pronounced at the 'tails' of the distributions, showing that, as expected, the hill-climbing type procedure is effective at improving the worst elements in the grid. Specifically, the minimum area-length metric is improved from $a_f = 0.67$ to $a_f = 0.94$, and the distribution of element-wise angles is narrowed from $31° \leq \theta_f \leq 112°$ to $44° \leq \theta_f \leq 78°$. A slight broadening of the mean parts of the distributions is also evident, showing that in some cases,
higher-quality elements are slightly compromised to facilitate improvements to their lower-quality neighbours. This behaviour is consistent with the *worst-first* philosophy employed in this study.

Beyond improvements to standard grid-quality metrics, the impact of mesh-optimisation can be further understood by considering the so-called *well-centredness* of the resulting staggered Voronoi/Delaunay dual grid. Well-centred triangulations are those for which all element circumcentres are located within their parent triangles, ensuring that the associated Voronoi cells
are *nicely-staggered* with respect to the underlying triangulation as a result. Such a constraint is equivalent to requiring that all Delaunay triangles are *acute*, such that $\theta_f \leq 90°$.

Well-centred grids are highly desirable from a numerical perspective, allowing, for instance, the mimetic-type C-grid discretisation scheme employed in the MPAS framework to achieve optimal rates of convergence. Specifically, when a grid is well-centred, it is guaranteed that associated edges in the staggered Voronoi and Delaunay cells intersect, ensuring that evalua-
tion of the element-wise transport and circulation terms can be accurately computed using local numerical stencils. In the case of *perfectly-centred* grids, such intersections occur at edge-midpoints — allowing a numerical scheme based on local linear interpolants to achieve fully second-order accuracy.

The construction of well-centred grids is known to be a difficult problem, and the development of algorithms for their generation is an ongoing area of research (VanderZee et al., 2008; Vanderzee et al., 2010). For the uniform resolution case





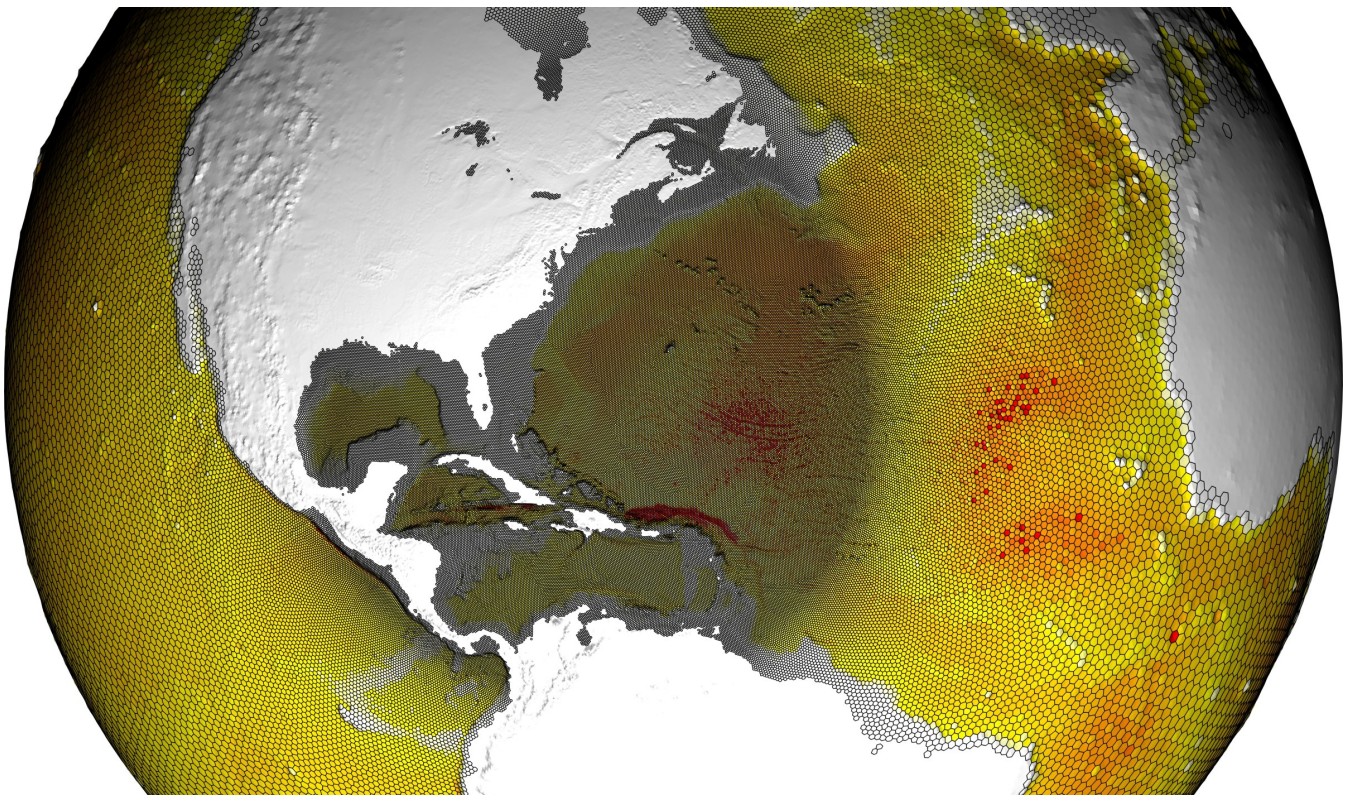

**Figure 7.** A regionally-refined Voronoi-type grid of the North Atlantic region. Global coarse grid resolution is $150\,\mathrm{km}$, with a $15\,\mathrm{km}$ eddy-permitting grid-spacing specified over the atlantic ocean basin. Topography is drawn using an exaggerated scale, with elevation from the reference geoid amplified by a factor of 10 in all cases.

studied here, it is clear that the hill-climbing type optimisation procedure is successful in generating a well-centred staggered Voronoi/Delaunay dual grid, with all enclosed angles less than $77.9°$.

### 4.3 Regionally-refined North Atlantic grid

The multi-resolution capabilities of the JIGSAW-GEO algorithm were investigated in the REGIONAL-ATLANTIC test-case,
5    seeking to build a regionally-resolved, staggered Voronoi/Delaunay-type dual grid for high-resolution modelling of the North Atlantic ocean basin. Non-uniform mesh-size constraints were enforced, setting $\bar{h}(\mathbf{x}) = 150\,\mathrm{km}$ globally, with $15\,\mathrm{km}$ eddy-permitting mesh-spacing specified over the north atlantic region. The resulting grid is shown in Figure 7 and contains 358,064 Delaunay triangles and 179,081 Voronoi cells. Grid-generation time was approximately $1\frac{1}{2}$ minutes, including both the initial restricted Frontal-Delaunay refinement and subsequent hill-climbing type optimisation. Grid-quality metrics are presented in
10    Figure 8, showing distributions before and after the application of the grid-optimisation procedure.



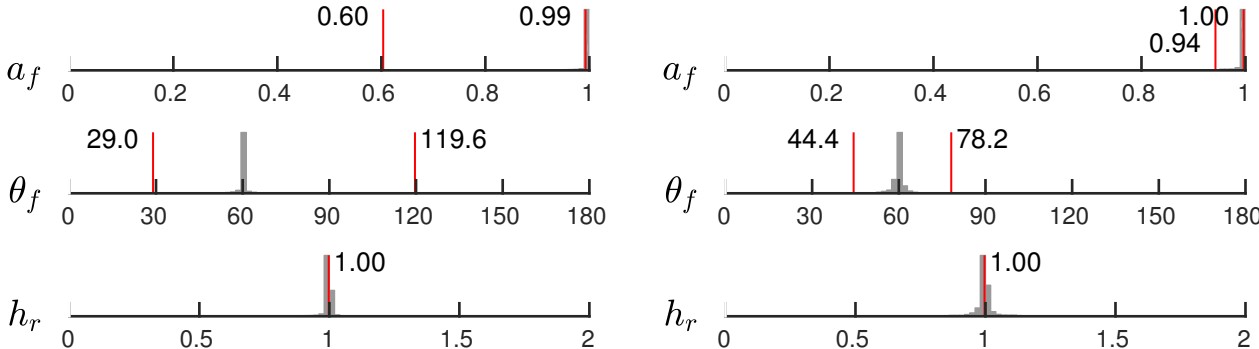

**Figure 8.** Mesh-quality metrics associated with the regionally-refined Voronoi-type grid of the North Atlantic region, before (left) and after (right) the application of hill-climbing mesh optimisation. Normalised histograms of element area-length ratio $a_f$, enclosed-angle $\theta_f$ and relative-length $h_r$ are illustrated, with minimum, maximum and mean values annotated.

Consistent with results presented previously, a very high-quality Voronoi/Delaunay grid was generated for the REGIONAL-ATLANTIC problem, with each grid-quality metric tightly clustered about its optimal value, such that $a_f \rightarrow 1$, $\theta_f \rightarrow 60°$ and $h_r \rightarrow 1$. The effect of the grid-optimisation procedure can be assessed by comparing the mesh-quality statistics presented in Figure 8. As per the uniform resolution test-case, mesh-optimisation appears to be most aggressive at the 'tails' of the distributions, acting to improve the worst elements in the grid. The minimum area-length metric is improved from $a_f = 0.60$ to $a_f = 0.94$, and the distribution of element-wise angles is narrowed from $29° \leq \theta_f \leq 120°$ to $44° \leq \theta_f \leq 78°$. The resulting optimised Voronoi/Delaunay staggered grid is also clearly *well-centred*, with all angles in the Delaunay trianglulation less than $78.2°$. Overall, grid-quality achieves essentially the same levels of optimality as the uniform resolution test-case, showing that the JIGSAW-GEO algorithm can be used to generate high-quality spatially-adaptive grids without obvious degradation in mesh-quality.

### 4.4 Multi-resolution Southern Ocean grid

The JIGSAW-GEO algorithm was then used to mesh the challenging SOUTHERN-OCEAN test-case, allowing its performance for large-scale problems involving rapidly-varying mesh-spacing constraints to be analysed in detail. This test-case seeks to build a multi-resolution, staggered Voronoi/Delaunay-type dual grid for regionally-refined ocean-modelling, with a particular focus on resolution of the Antarctic Circumpolar Current (ACC), and adjacent Antarctic processes. Composite mesh-spacing constraints were enforced, consisting of a coarse global background resolution of $150\,\mathrm{km}$, with an *eddy-permitting* $15\,\mathrm{km}$ grid-spacing specified south of $32.5°\mathrm{S}$. Additional topographic adaptation is also utilised in the southern annulus region, with grid-resolution increased in regions of large bathymetric gradient. A minimum grid-spacing of $4\,\mathrm{km}$ was specified. Topographic gradients were computed using the high-resolution ETOPO1 Global Relief dataset (Amante and Eakins, 2009). The resulting grid is shown in Figure 11, with additional detail shown in Figure 11. The grid contains 3,119,849 Delaunay triangles and 1,559,927 Voronoi cells. Grid-generation time was approximately 10 minutes, including both the initial restricted Frontal-





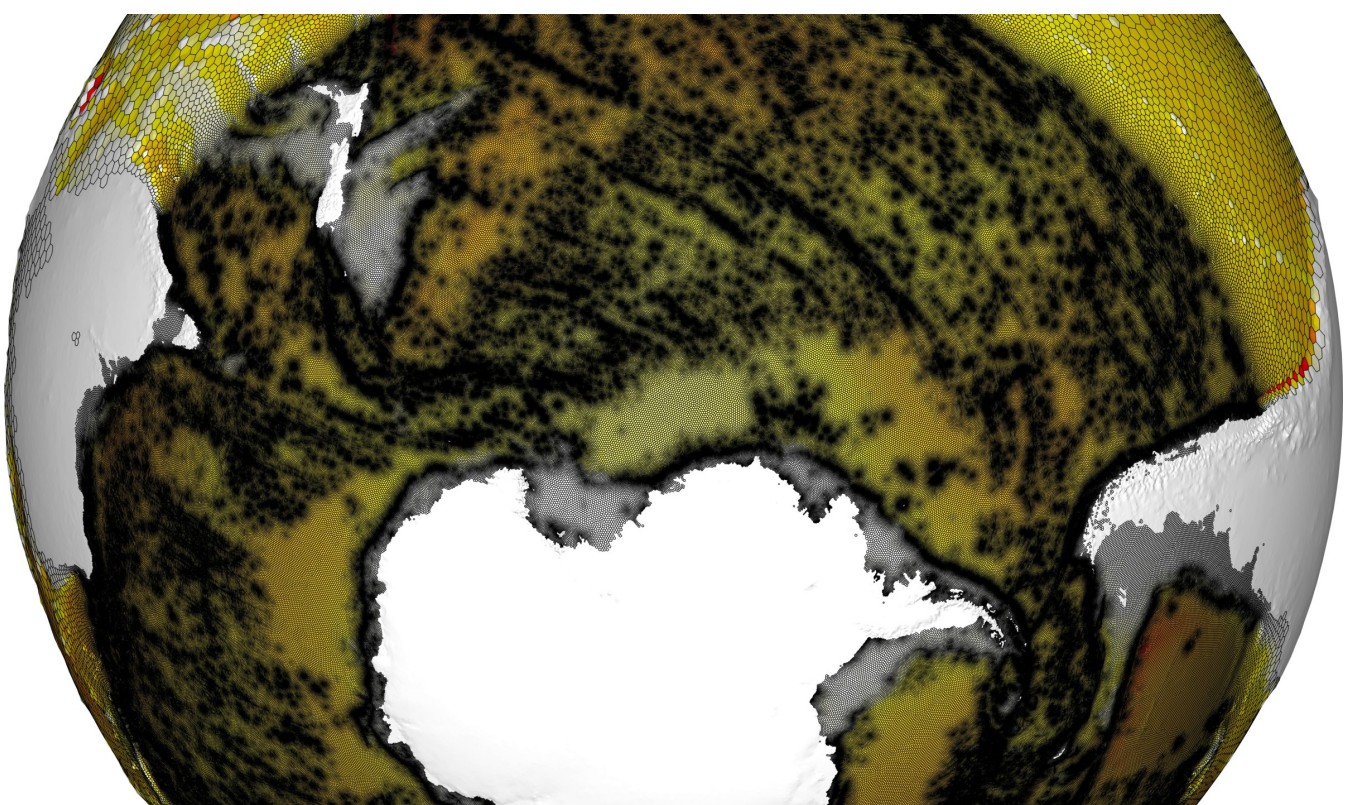

**Figure 9.** A multi-resolution Voronoi-type grid of the Southern Ocean. Global coarse grid resolution is $150\,\mathrm{km}$, with a $15\,\mathrm{km}$ eddy-permitting grid-spacing specified south of $32.5°\,\mathrm{S}$. Additional topographic adaptation is also utilised in the southern annulus region, with grid-resolution increased in areas of large bathymetric gradient. Minimum grid-spacing is $4\,\mathrm{km}$. Topography is drawn using an exaggerated scale, with elevation from the reference geoid amplified by a factor of 10 in all cases.

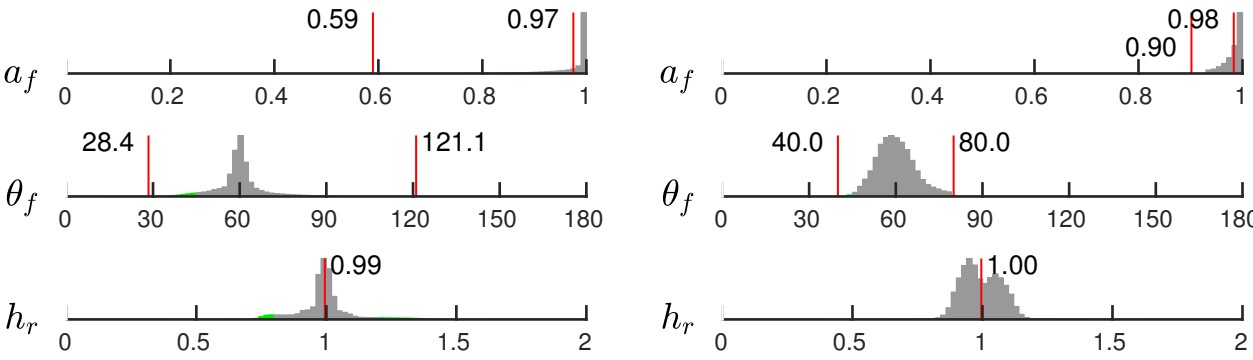

**Figure 10.** Mesh-quality metrics associated with the multi-resolution Voronoi-type grid of the Southern Ocean, before (left) and after (right) the application of hill-climbing mesh optimisation. Normalised histograms of element area-length ratio $a_f$, enclosed-angle $\theta_f$ and relative-length $h_r$ are illustrated, with minimum, maximum and mean values annotated.



Delaunay refinement and subsequent hill-climbing type optimisation stages. The grid-optimisation phase required approximately four times the computational effort of the initial Delaunay refinement. Associated grid-quality metrics are presented in Figure 10, showing distributions before and after the application of the grid-optimisation procedure.

Consistent with the uniform resolution test-case presented previously, visual inspection of Figures 9 and 11 confirm that the JIGSAW-GEO algorithm is capable of generating very high quality multi-resolution grids, containing a majority of near-perfect Delaunay triangles and Voronoi cells. Additionally, it can be seen that grid resolution varies smoothly, even in regions of rapidly-fluctuating mesh-spacing constraints, as per the topographically induced refinement patterns shown in Figure 11. Analysis of the grid-quality metrics shown in Figure 10 shows that very high levels of mesh regularity are achieved, with element area-length scores tightly clustered about $a_f = 1$ and element-angles showing strong convergence around $\theta_f = 60°$. Interestingly, despite the complexity of the imposed mesh-spacing function, analysis of the relative-length distribution still shows relatively tight conformance, with a sharp clustering about $h_r = 1$. Overall, mean grid-quality is slightly reduced compared to the uniform resolution case, illustrated by a slight broadening of the grid-quality distributions. Note that such behaviour is expected in the multi-resolution case, with slightly imperfect triangle geometries required to satisfy the non-uniform mesh-spacing constraints. The minimum enclosed-angle in the un-optimised grid can also be seen to lie exactly at the lower bound of $28.4°$.

The effect of the grid-optimisation procedure can be assessed by comparing the mesh-quality statistics presented in Figure 10. As per the uniform resolution test-case, mesh-optimisation appears to be most aggressive at the 'tails' of the distributions, acting to improve the worst elements in the grid. The minimum area-length metric is improved from $a_f = 0.59$ to $a_f = 0.90$, and the distribution of element-wise angles is narrowed from $28° \leq \theta_f \leq 121°$ to $40° \leq \theta_f \leq 80°$. Consistent with previous results, a slight broadening of the mean components of the distributions can be observed, especially in the enclosed-angle and relative-length metrics. This behaviour shows that, in this case, improvements to worst-case grid-quality are achieved through slight compromises to mean-quality and mesh-spacing conformance. Note that the resulting optimised Voronoi/Delaunay staggered grid is also *well-centred*, with all angles in the Delaunay trianglulation less than $80°$. This result shows only a marginal degradation compared to the uniform resolution example presented previously — despite the complexity of the imposed grid-spacing function. This result demonstrates the effectiveness of the optimisation strategies presented here, and shows that very high-quality, well-centred grids can be generated even for general multi-resolution cases. Nonetheless, the construction of well-centred grids remains a challenging task, and it is expected that it may be possible to design a test-case that defeats the current strategy. As such, the pursuit of alternative mesh optimisation strategies, designed to target grid well-centredness directly, is an interesting avenue for future research.

# 5 Conclusions & Future Work

A new algorithm for the generation of non-uniform, locally-orthogonal staggered unstructured grids for large-scale general circulation modelling has been described. Using a combination of Frontal-Delaunay refinement and hill-climbing type optimisation, it has been shown that very high-quality Voronoi/Delaunay type staggered grids can be generated for geophysical





**Figure 11.** A multi-resolution Voronoi-type grid of the Southern Ocean. Global coarse grid resolution is $150\,\mathrm{km}$, with a $15\,\mathrm{km}$ eddy-permitting grid-spacing specified south of $32.5°\,\mathrm{S}$. Additional topographic adaptation is also utilised in the southern annulus region, with grid-resolution increased in areas of large bathymetric gradient. Minimum grid-spacing is $4\,\mathrm{km}$. Topography is drawn using an exaggerated scale, with elevation from the reference geoid amplified by a factor of 10 in all cases.





applications on the sphere, with a focus on global atmospheric and oceanic type modelling. These new algorithms are available as part of the JIGSAW meshing package, providing a simple and easy-to-use tool for the oceanic and atmospheric modelling communities. A number of global-scale benchmark problems have been analysed, verifying the performance of the new approach, and demonstrating that very high-quality meshes can be generated for large-scale global problems, including those

incorporating highly non-uniform mesh-spacing constraints. The Frontal-Delaunay refinement algorithm has been shown to generate *guaranteed-quality* spheroidal Delaunay triangulations — satisfying worst-case bounds on element-wise angles and exhibiting smooth grading characteristics. The use of a coupled geometrical and topological hill-climbing type optimisation procedure was shown to further improve mesh-quality statistics, especially for the lowest quality elements in each mesh. It was demonstrated that these optimisation techniques allow grid-quality to be improved to the extent that fully *well-centred* mesh

configurations are achieved, with all angles in the surface triangulation bounded below $90°$. For the three global test-cases presented here, enclosed-angles in the range $40° \leq \theta_f \leq 80°$ were achieved.

    The construction of locally-orthogonal staggered polygonal grids, appropriate for a range of contemporary unstructured general circulation models, has been discussed in detail. Future work will focus on a generalisation of the algorithm and improvements to its efficiency, including: (i) support for inscribed geometrical constraints, such as coastlines, (ii) the use

of multi-threaded programming patterns to improve computational performance, and (iii) further enhancements to the mesh optimisation procedure, with a focus on improving the *well-centredness* of the resulting staggered grids. The investigation of *solution-adaptive* multi-scale representations, in which grid-resolution is adapted to spatial variability in model state (Sein et al., 2016), is also an obvious direction for future investigation.

*Acknowledgements.*  This work was carried out at the NASA Goddard Institute for Space Studies, the Massachusetts Institute of Technology,

and the University of Sydney with the support of a NASA–MIT cooperative agreement and an Australian Postgraduate Award. The author wishes to thank John Marshall and Todd Ringler for their comments on an earlier version of the manuscript.

## 6   Code availability

The JIGSAW-GEO grid-generator can be found online at github.com/dengwirda/jigsaw-geo-matlab

## Appendix A: Spheroidal Predicates

Recalling the methodology described in Section 2, computation of the *restricted* Delaunay surface tessellation $\mathrm{Del}|_\Sigma(X)$ requires the evaluation of a single geometric predicate. Given a spheroidal surface $\Sigma$, the task is to compute intersections between edges in the Voronoi tessellation $\mathrm{Vor}(X)$ and the input features $\Sigma$.





## A1 Restricted surface triangles

Restricted surface triangles $f_i \in \mathrm{Del}|_\Sigma(X)$ are defined as those associated with an *intersecting* Voronoi edge $v_e \in \mathrm{Vor}(X)$, where $v_e \cap \Sigma \neq \emptyset$. These triangles provide a good piecewise linear approximation to the surface $\Sigma$. For a given triangle $f_i$, the associated Voronoi edge $v_e$ is defined as the line-segment joining the two *circumcentres* $\mathbf{c}_i$ and $\mathbf{c}_j$ associated with the pair of tetrahedrons that share the face $f_i$. The task then is to find intersections between the line-segments $v_e$ and the surface $\Sigma$.

Let $\mathbf{p}$ be a point on a given Voronoi edge-segment $v_e$

$$\mathbf{p} = \bar{\mathbf{c}} + t\Delta, \quad -1 \leq t \leq +1, \qquad \text{where:} \quad \bar{\mathbf{c}} = \tfrac{1}{2}(\mathbf{c}_i + \mathbf{c_j}), \quad \Delta = \tfrac{1}{2}(\mathbf{c}_j - \mathbf{c_i}). \tag{A1}$$

Substituting (A1) into the expression for the spheroidal surface (1), the existence of real, bounded solutions, such that $-1 \leq t \leq +1$, indicates a non-trival intersection $v_e \cap \Sigma \neq \emptyset$. Specifically, expanding and rearranging after substitution

$$\sum_{i=1}^{3}\left(\frac{\bar{\mathbf{c}}_i + t\Delta_i}{r_i}\right)^2 = 1, \qquad \sum_{i=1}^{3}\frac{\bar{\mathbf{c}}_i^2 + 2t\bar{\mathbf{c}}_i\Delta_i + t^2\Delta_i^2}{r_i^2} = 1, \qquad \sum_{i=1}^{3}\left(\frac{\Delta_i^2}{r_i^2}\right)t^2 + \left(\frac{2\bar{\mathbf{c}}_i\Delta_i}{r_i^2}\right)t + \left(\frac{\bar{\mathbf{c}}_i^2}{r_i^2} - 1\right) = 0, \tag{A2}$$

which is simply a quadratic expression for the parameter $t$ and can be solved using the standard approach. Given a real solution $-1 \leq t_\Sigma \leq +1$ the corresponding point of intersection $\mathbf{p}_\Sigma$ can be found using (A1).



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
