# Peer review of "JIGSAW-GEO (1.0): Locally-orthogonal staggered unstructured grid-generation for general circulation modelling on the sphere\"

_Geoscientific Model Development, 2016_

## Short Comment (SC1) · 3 Jan 2017

Dear authors,

in my role as Executive editor of GMD, I would like to bring to your attention our Editorial version 1.1:

http://www.geosci-model-dev.net/8/3487/2015/gmd-8-3487-2015.html

This highlights some requirements of papers published in GMD, which is also available on the GMD website in the 'Manuscript Types' section:

http://www.geoscientific-model-development.net/submission/manuscript_types.html

In particular, please note that for your paper, the following requirement has not been met in the Discussions paper:

- "The main paper must give the model name and version number (or other unique identifier) in the title."

In order to simplify reference to your developments, please add the name of the grid-generator "JIGSAW-GEO grid-generator" and a version number in the title of your article in your revised submission to GMD.

Yours,

Astrid Kerkweg

—————————————————————

---

## Referee Comment (RC1) · Anonymous Referee #1 · 9 Jan 2017

The current paper deals with high quality surface triangulation applied to general circulation modelling. The paper bypasses a parametric representation of an arbitrary surface by limiting the surface definition to an ellipsoid representing the earth. The main algorithm relies on a coupled Frontal-Delaunay approach. Various examples are provided to illustrate the method.

Overall, the paper is clear and there is obviously a lot of work in it. However, my main critic is that there is not much new brought by the paper, as opposed to what is claimed in the conclusion, except for the fact of applying it to a general circulation modelling. The curvature of the Earth is almost constant so technically it is not difficult to surface mesh it. The paper introduces a lot of concepts such as restricted Delaunay, Frontal-

Delaunay, Hill-climbing, Gradient based smoothing, etc, which are very classic and well established unstructured mesh techniques. At least, it should be clearly stated what is new.

- The claim that Voronoi edges are always perpendicular to mesh edges is wrong. It is only valid for an acute triangulation, which is not true in general, particularly because of the boundaries. This is a property that has been pursued by the electromagnetic solvers for a long time for the same reason but only partially reached. This is only briefly mentioned in Section 4.

- line 28. It is not clear at all in general that maximisation of the minimum angle is beneficial. Add references.

- The pictures do not clearly show the mesh transitions for size variations in details.

- The abstract mentions a-priori guaranteed quality bounds while nothing is proved. Empirical studies show good results but no bounds are provided.

- The abstract is misleading. The code may be recently developped but the techniques used in it are not recent.

- There is not detail about the initialization on the sphere of the algorithm. You mention that the algorithm scans the triangles that do not verify given criteria, but how are these initial triangles created?

―――――――――――――――――――――――

---

## Referee Comment (RC2) · Anonymous Referee #1 · 11 Jan 2017

I thank the author for his answers. The author clearly answered to my previous questions. I only have a few issues left.

1 Yes you are right about Voronoi-Delaunay grid staggering. I was too fast and apologize. I indeed was refering to what you call well-centered.

2 My main concern is still about the novelty of the method. You say:

In the current work, it's shown that a combination of Frontal-Delaunay refinement and hill-climbing optimisation is an effective strategy — able to produce very high-quality well-centred Voronoi-Delaunay grids even when complex, highly non-uniform grid sizing constraints are imposed. I believe this to be a new result of benefit to the unstruc-

tured oceanic/atmospheric modelling communities. Public availability of the associated JIGSAW-GEO grid-generator is also thought to be a further benefit to the community.

and:

I do not believe that the methods presented in the present work are preexisting. The hybrid Frontal-Delaunay surface meshing technique described here, able to guarantee worst-case bounds on element quality and sizing conformance are, in my view, new. I am not aware of another algorithm with the same properties — able to produce smoothly varying Voronoi-Delaunay grids with very high mean element quality (similar to advancing front type schemes), while also guaranteeing worst-case bounds on element angles and conformance (a'la standard Delaunay-refinement techniques). Existing methods for unstructured oceanic/atmospheric modelling appear to either lack provable worst-case bounds [Jacobsen et al., 2013], or generally produce grids with somewhat lower overall quality [Lambrechts et al., 2008]. The combination of the Frontal-Delaunay scheme with a coupled hill-climbing optimisation strategy to generate 'well-centred' grids is also, in my view, new.

EVERY mesh generator (edge, face, volume) has a main engine (Delaunay, Frontal, Octree, coupled) and an optimization phase that follows [1], so there is nothing new to that. The facts that you apply it to oceanic/atmospheric communities or that it is publicly available do not make these techniques new. I have added a list of references on high quality surface mesh generation that present the same high quality based on the same techniques [2,3,4,5] on top of the coupled Delaunay advancing front variants which are not fully referenced. Feel free to include them or not.

Nevertheless, I completely agree with the fact that the application of these techniques to the oceanic/atmospheric communities is new and interesting and therefore, the paper should be published.

[1] Frey,P.J. and George,P.L., Meshing, applications to finite elements, Hermes, Paris, 1999 [2] J. Tristano, S. Owen, S. Canann, Advancing front surface mesh generation

in parametric space using a Riemannian surface definition, in: IMR, 1998, pp. 429–445. [3] D. Rypl, P. Krysl, Triangulation of 3D surfaces, Eng. Comput. 13 (1997) 87–98. [4] C. Lee, Automatic metric advancing front triangulation over curved surfaces, Eng. Comput. 17 (1) (2000) 48–74. 642–667. [5] Löhner R. Regridding surface triangulations. Journal of Computational Physics 1996; 126:1–10.

---

## Referee Comment (RC3) · Anonymous Referee #2 · 19 Jan 2017

The manuscript is of interest, but I see several points for improvement.

There is a sufficiently detailed description of the techniques used to improve the mesh quality, but the frontal-Delaunay technique is just mentioned. It would be helpful if it is described in some detail, for the intention of this paper is to help the potential user to learn about the technical details of the algorithm, and it is still a bit difficult to do. It would be very helpful to present a schematic of the algorithm at the very beginning, followed by the description of separate steps. This is partly done for the iterative mesh quality improvement, but I think that the entire algorithm has to be included. Also I would advice to be more clear with what is new in the algorithm.

Editorial remarks:

Fig 1. The coloring used is non-informative. I was struggling to see some familiar topographic features but I could not. I would recommend to omit the topographic height, it does not have any sense here, only distracts you reader. Same concerns other mesh figures.

Page 2, line 6 'A majority of ...' — I do not think this statement is correct. Many of practically used ocean circulation models use so-called tripolar meshes with Mercator-type stretching. Cubed sphere is used less frequently (the internal Rossby radius of deformation in the ocean is decreasing toward high latitudes, and meshes refined in high latitudes are more natural)

Line 10 'leading to significant...' If lon-lat mesh is used for ocean modeling, it is of course rotated, so that poles are on the land and there is no singularity. What is discussed has relevance to the atmosphere, but not to the ocean. Since your mesh examples are related to the ocean, the discussion creates misunderstanding.

Page 3,

line 5 The discussion here misses the point that FESOM, FVCOM, SLIM, Fluidity may work on general triangular meshes. SUNTANS (and its numerous predecessors) need orthogonal (well-centered) meshes (with the circumcenters inside respective triangles).

Line 29 I think the main point of Lambrecht et al. is that a care is taken of the shape of coastlines, resolution of passages etc. This question is not reflected in the manuscript (except for conclusions), although it presents the main challenge. It is quality of triangles close to coastlines that is problematic in many practical cases.

I would also recommend to mention Admesh (DOI 10.1007/s10236-012-0574-0) which relies on the Persson's approach.

Page 4. Section 2. I would start here with brief description of the entire algorithm. Your reader keeps wondering what is the algorithm before the end of section 3. Present details of the Frontal -Delaunay algorithm and explain that in reality it is a 3D procedure

with the central point of the sphere used to form the tetrahedra, and the restriction is just surface triangulation. Otherwise your preliminaries and Fig.2 are a bit embarrassing for a general reader (who would keep asking about v_e and its relation to the surface).

Page 5 Line 13 Restricted Delaunay tesselation — try to explain this better, by using illustrations. I do not see your Fig. 2 to be of much help.

It is not clear how mesh spacing functions are used in the mesh construction. Are they taken in account in the frontal procedure? It is not mentioned. Help you reader to clearly see the steps of the algorithm.

Page 7.

Line 10 .. fully inverted elements? Please define what do you mean.

Fig. 3. Is not it generally known?

Page 8 Line 16 Fluid velocity and vorticity are commonly at different locations.

Page 9, Line 4 ... Mention that you mean C-grid type techniques. There are other possibilities.

line 7 'described previously'? It was not really described here.

Beginning from line 9, there is discussion that is either well known or irrelevant to the mesh generation.Why do you need it?

Page 11.

Please define all quantities in (7) and better explain how computations are implemented.

line 14 What is the grid-quality vector?

line 26 What is the lexicographical comparison?

Section 5. Please clearly define the novelty. What is describe is the selection of known steps.

page 22 , line 14 (ii)–? I think it is secondary and technical issue. Well-centeredness and coastlines are real algorithmic queastions.

———————————————

---

## Author Response (AR1)

To: The Topical Editor
Geoscientific Model Development

Dear Editor,

Please find enclosed a revision of the manuscript: '**JIGSAW-GEO (1.0): Locally-orthogonal staggered unstructured grid-generation for general circulation modelling on the sphere**' to be considered for publication in *Geoscientific Model Development*.

I have appended here a detailed response to the initial reviews of the paper, including a marked-up manuscript-diff and a detailed response to all comments from the reviewers. I have revised the manuscript according to the changes I have noted in-line in my response to the reviews.

Sincerely,

Darren Engwirda

Postdoctoral Associate
Dept. of Earth, Atmospheric and Planetary Science
Massachusetts Institute of Technology
engwirda [at] mit [dot] edu

**JIGSAW-GEO (1.0): Locally-orthogonal staggered unstructured grid-generation for general circulation modelling on the sphere[*]**

Darren Engwirda[1,2]

[1]Department of Earth, Atmospheric and Planetary Sciences, Room 54-1517, Massachusetts Institute of Technology, 77 Massachusetts Avenue, Cambridge, MA 02139-4307
[2]NASA Goddard Institute for Space Studies, 2880 Broadway, New York, NY 10025 USA

*Correspondence to:* Darren Engwirda (engwirda@mit.edu)

**Abstract.** An algorithm for the generation of non-uniform, locally-orthogonal staggered unstructured spheroidal grids is described. This technique is designed to generate  high-quality staggered Voronoi/Delaunay  meshes appropriate for general circulation modelling on the sphere, including applications to atmospheric simulation, ocean-modelling and numerical weather prediction. Using a recently developed Frontal-Delaunay refinement technique, a method for the construction of high-quality unstructured spheroidal Delaunay triangulations is introduced. A locally-orthogonal polygonal grid, derived from the associated Voronoi diagram, is computed as the staggered dual. It is shown that use of the  Frontal-Delaunay refinement technique allows for the generation of  very high-quality unstructured triangulations, satisfying a-priori bounds on element size and shape. Grid-quality is further improved through the application of hill-climbing type optimisation techniques.  Overall, the algorithm is shown to produce grids with very high element quality and smooth grading characteristics, while imposing relatively low computational expense.  A selection of uniform and non-uniform spheroidal grids appropriate for high-resolution, multi-scale general circulation modelling are presented. These grids are shown to satisfy the geometric constraints associated with contemporary unstructured C-grid type finite-volume models, including the Model for Prediction Across Scales (MPAS-O). The use of user-defined mesh-spacing functions to generate smoothly graded, non-uniform grids for multi-resolution type studies is discussed in detail.

**Keywords.** Grid-generation; Frontal-Delaunay refinement; Voronoi tessellation; Grid-optimisation; Geophysical fluid dynamics; Ocean modelling; Atmospheric modelling; Numerical weather predication; Model for Prediction Across Scales (MPAS)

**1 Introduction**

The development of atmospheric and oceanic general circulation models based on *unstructured* numerical discretisation schemes is an emerging area of research. This trend necessitates the development of unstructured grid-generation algorithms  designed to produce very high-resolution, *guaranteed-quality* unstructured triangular and polygonal
* * *
[*]A short version of this paper appears in the proceedings of the 24th International Meshing Roundtable (**?**).

[Figure]

**Figure 1.** Conventional semi-structured meshing for the sphere, showing a regular cubed-sphere type grid (left), and a regular icosahedral class grid (right). Both grids were generated using equivalent target mean edge lengths, and are coloured according to mean topographic height at grid-cell centres. Topography is drawn using an exaggerated scale, with elevation from the reference geoid amplified by a factor of 20 in both cases.

meshes that satisfy non-uniform mesh-spacing distributions and embedded geometrical constraints. This study investigates the applicability of a recently developed surface meshing algorithm (**??**) based on restricted Frontal-Delaunay refinement and hill-climbing type optimisation for this task.

**1.1 Semi-structured grids**

5   While simple structured grid types for the sphere can be obtained by assembling a uniform discretisation in spherical coordinates, the resulting *lat-lon* grid is often inappropriate for numerical simulation, due to the presence of strong *grid-singularities* at the two poles. Such features manifest as local distortions in grid-quality, consisting of regions of highly distorted quadrilateral grid-cells. These low-quality elements can lead to a number of undesirable numerical effects — imposing restrictions on model time-step and stability, and compromising local spatial accuracy.  As a result, a majority of current generation general

10   circulation models  are instead based on  *semi-structured* quadrilateral discretisation schemes, including the *cubed-sphere*  (**???**) and *tri-polar* type configurations (**???**).

In the cubed-sphere framework, the spherical surface is decomposed into a cube-like topology, with each of  the six quadrilateral faces discretised as a structured curvilinear grid. In such an arrangement, the two strong grid-singularities of the lat-lon configuration are replaced by eight weak discontinuities at the cube corners, leading to significant improvements in

numerical performance. **?** present detailed discussions of techniques for the generation and optimisation of cube-sphere type grids. A regular *gnomonic-type* cubed-sphere grid is illustrated in Figure 1a. In the tri-polar grid, the present-day continental configuration is exploited to *bury* the singularities associated with a three-way polar decomposition of the sphere outside of the ocean mask. The resulting *numerically-active* subset of the grid is well-conditioned as a result. While such configurations are a popular choice for models designed for present-day Earth-based ocean studies, the generality of these methods is clearly limited. In this study, we instead pursue the development of more general-purpose techniques, applicable for both ocean and atmospheric modelling in general planetary, present-day and paleo-Earth environments.

In addition to the standard cubed-sphere and tri-polar type configurations, a second class of semi-structured spherical grid can be  constructed through *icosahedral-type* decompositions (**??**). In such cases, the primary grid is defined as a regular spherical triangulation, obtained through recursive bisection of the icosahedral configuration. Additionally, a staggered polygonal *dual* grid, consisting of hexagonal and pentagonal cells, is often used as a basis for finite-volume type numerical schemes. This geometrical duality is an example of the  *locally-orthogonal* Delaunay/Voronoi type grid staggering that forms the basis of this paper. Icosahedral-type grids provide a near-perfect tessellation of the sphere — free of topological discontinuities and/or geometric irregularity. Such methods are applicable to both atmospheric and oceanic type simulations. A regular icosahedral-class grid is illustrated in Figure 1b.

**1.2  Unstructured grids**

While the semi-structured grids described previously each provide effective frameworks for uniform resolution global simulation, the development of *multi-resolution* modelling environments requires alternative techniques. A range of new general  circulation models, including the Finite Element Sea Ice-Ocean Model (FESOM) (**?**), the Finite Volume Community Ocean Model (FVCOM) (**???**), the Stanford Unstructured Non-hydrostatic Terrain-following Adaptive Navier-Stokes Simulator (SUNTANS) (**??**), and the Second-generation Louvain-la-Neuve Ice-ocean Model (SLIM) (**??**) are based on semi-structured triangular grids, with the horizontal directions discretised according to an unstructured spherical triangulation, and the vertical direction represented as a stack of locally structured layers. The Model for Predication Across Scales (MPAS) (**???**) adopts a similar arrangement, except that a *locally-orthogonal* unstructured discretisation is adopted, consisting of both a Spherical Voronoi Tessellation (SVT) and its dual Delaunay triangulation. The use of fully unstructured representations, based on general tetrahedral and/or polyhedral grids, are also under investigation in the Fluidity framework (**????**). Such models all impose different requirements on the *quality* of the underlying unstructured grids, with some models, including FESOM, SLIM and Fluidity, offering additionally flexibility. In all cases though, the performance of the numerical simulation can be expected to improve with increased grid-quality — encouraging the search for optimised grid-generation algorithms. A full discussion of grid-quality constraints for general circulation modelling is presented in Section 2.

Existing approaches for unstructured grid-generation on spherical geometries have focused on a number of techniques, including: (i) the use of iterative,  optimisation-type algorithms designed to construct Spherical Centroidal

Voronoi Tessellations (SCVT's) (**?**), and (ii) the adaptation of anisotropic two-dimensional meshing techniques (**?**) that build grids in associated parametric spaces. The MPI-SCVT algorithm (**?**) is a massively parallel implementation of iterative Lloyd-type smoothing (**?**) for the construction of SVCT's for use in the MPAS framework. In this approach, a set of vertices are distributed over the spherical surface and iteratively *smoothed* until a high-quality Voronoi tessellation is obtained. Specifically, each iteration repositions vertices to the centroids of their associated Voronoi cells and updates the topology of the underlying spherical Delaunay triangulation. While such an approach typically leads to the generation of high-quality *centroidal* Voronoi tessellations on the sphere (SCVT's), the algorithm does not provide theoretical guarantees on minimum element quality, and often requires significant computational effort to achieve convergence. Additionally, current implementations of the MPI-SCVT algorithm do not provide a mechanism to constrain the grid to embedded features, such as coastal boundaries.

**?** present an unstructured spherical triangulation framework using the general-purpose grid-generation package Gmsh (**?**). In this work, unstructured spherical triangulations are generated for the world ocean using a parametric meshing approach. Specifically, a triangulation of the spherical surface is generated by *mapping* the full domain (including coastlines) on to an associated two-dimensional parametric space via stereographic projection. Importantly, as a result of the projection, the grids constructed in parametric space must be highly anisotropic, such that a well-shaped, isotropic triangulation is induced on the sphere. A range of existing two-dimensional anisotropic meshing algorithms are investigated, including Delaunay-refinement, advancing-front, and adaptation-type approaches. In particular, the algorithm is designed to ensure a faithful representation of complex coastal boundary conditions. While a detailed model of such constraints is often neglected in global simulations, resolution of these features is a key factor for regional and coastal models. Several algorithms support the generation of unstructured two-dimensional grids for such domains, including the ADmesh package (**?**) and the Stomel library (**?**).

The current study explores the development of  a new algorithm for the generation of high-resolution, guaranteed-quality spheroidal Delaunay triangulations and associated Voronoi tessellations — appropriate for a range of  unstructured-grid type general circulation models. In this work, meshes are generated on the spheroidal surface directly, without need for local parameterisation or projection. Such an approach will be shown to exhibit significant flexibility — immune to issues of coordinate singularity and/or continental configuration. The applicability of this approach to grid-generation for imperfect spheres, including oblate spheroids and general ellipsoidal surfaces is  also explored. Overall, significant effort is invested to develop techniques designed to produce very high-quality multi-resolution grids appropriate for contemporary unstructured C-grid type models, such as the MPAS framework. The paper is organised as follows: an overview of grid-generation for general circulation modelling is presented in Section 2, outlining various constraints and minimum requirements on grid-quality. A description of the Frontal-Delaunay refinement and hill-climbing type optimisation algorithms is given in Sections 3 and 4. A set of uniform and non-uniform grids appropriate for high-resolution, multi-scale general circulation modelling are presented Section 5, alongside an analysis of computational performance and optimality. Avenues for future work are outlined in Section 6.

[Figure]

**Figure 2.** Construction of the staggered surface Voronoi control-volumes, illustrating: (left) locally-orthogonal Voronoi/Delaunay staggering, and (right) an associated unstructured C-grid type numerical discretisation scheme, as per the MPAS model. The formulation is a combination of conservative *cell-centred* tracer quantities $\bar{\psi}_i$, *edge-centred* normal velocity components $(\mathbf{u} \cdot \hat{\mathbf{n}})_j$, and auxiliary *vertex-centred* vorticity variables $\xi_k$.

**2 Grid-generation for general-circulation modelling**

Numerical formulations for large-scale atmospheric and/or oceanic general circulation modelling are often based on a *staggered* grid configuration, with quantities such as fluid pressure, geopotential, and density discretised using a primary control-volume, and the fluid velocity field and vorticity distribution represented at secondary, spatially distinct grid-points. In the context of standard structured grid types, various staggered arrangements are described by the well-known Arakawa schemes (**?**).

The development of general circulation models based on unstructured grid types is an emerging area of research, and, as a result, a variety of numerical formulations are currently under investigation. In this study, the development of *locally-orthogonal* grids appropriate for staggered unstructured C-grid type numerical schemes are pursued, as these methods are thought to represent the most logical extension of the conventional structured Arakawa-type techniques to the unstructured setting. Such formulations require that grids satisfy a *local-orthogonality* constraint, with adjacent grid-cell edges in the primary and secondary control-volumes required to be mutually perpendicular. In the unstructured setting, it is known that the Delaunay triangulation and Voronoi tessellation constitute a locally-orthogonal staggered dual, leading to a natural framework for the construction of such unstructured meshes.

Consisting of a set of (convex) polygonal grid-cells centred on each vertex in the underlying triangulation, the surface Voronoi diagram $\mathrm{Vor}|_\Sigma(X)$ obeys a number of local orthogonality constraints. Specifically, grid-cell edges in the Voronoi tessellation are guaranteed to be perpendicular to their associated edges in the underlying Delaunay triangulation, passing

[Figure]

**Figure 3.** Comparison of *well-centred* and poorly-staggered Voronoi/Delaunay dual grids, from left to right, respectively. In the well-centred configuration, each vertex of the Voronoi polygon lies within the interior of its associated Delaunay triangle. A conservative computation of the vertex-centred vorticity variables can therefore be achieved using the three velocity components adjacent to each circumcentre. In the poorly-staggered configuration, one Voronoi vertex lies outside its associated triangle (shaded). The associated Voronoi and Delaunay edges do not intersect as a result, and the triangle-based vorticity reconstruction is no longer valid. Note that the quality of the triangulation here is not pathological, with all angles bounded below $\theta_f \lesssim 120°$. The construction of fully well-centred grids can be seen as a difficult problem as a result, requiring the assembly of very high-quality Delaunay triangulations.

through the Delaunay-edge midpoints. Additionally, in the case of perfectly *regular* and *centroidal* tessellations, the Delaunay-edges are guaranteed to pass through the midpoints of their associated Voronoi duals. Voronoi grid-cells are formed as the convex-hull of the incident element *circumcentres* associated with the set of surface triangles adjacent to each vertex. Example Voronoi/Delaunay type unstructured grid staggering is illustrated in Figure 2.

5    While detailed comparisons of particular numerical discretisation schemes lie outside the scope of the current study, brief comments regarding the benefits of locally-orthogonal grid-staggering arrangements are made. Pursuing an unstructured variant of the widely-used Arakawa C-grid, the placement of fluid pressure, geopotential and density degrees-of-freedom within the primary Voronoi control-volumes, and orthogonal velocity vectors on Delaunay-edges achieves a similar configuration. Such an arrangement facilitates construction of a standard conservative finite-volume type scheme for the transport of fluid properties

10  and a *mimetic* class (**??**) finite-difference formulation for the evolution of velocity components. Additionally, exploiting alignment with Delaunay-edges, a conservative evaluation of the fluid vorticity can be made on the staggered Delaunay triangles. Overall, this scheme is known to posses a variety of desirable conservation properties, conserving mass, potential vorticity and enstrophy, and preserving geostrophic balance. This *unstructured* C-grid scheme is currently employed in the Model for Prediction Across Scales (MPAS) for both atmospheric and oceanic modelling (**???**). See Figure 2 for additional

15  details.

While numerically elegant, such unstructured C-grid schemes impose a heavy-burden on the quality of the underlying unstructured grid, requiring not only that grids be locally-orthogonal, but also *well-centred* and mutually *centroidal*. The *well-centred*-ness and *centroidal*-ness of an unstructured grid are constraints related to the nature of the staggering between the primary and secondary grid-cells. Specifically, a grid is *well-centred* when all dual Voronoi vertices lie within the interior of their associated Delaunay triangles. Such a constraint guarantees that adjacent Delaunay and Voronoi edges intersect, and, in the context of the unstructured C-grid scheme described previously, guarantees that a consistent stencil exists for the reconstruction of the discrete vorticity variable. In practice, the construction of well-centred grids is known to be particularly onerous (**??**), requiring that the triangulation consist of all-acute elements. See Figure 3 for additional detail.

Unstructured grids are *centroidal* when their primary and secondary vertices lie at the centres-of-mass of their associated dual grid-cells, with the vertices of the Voronoi polygons lying at the centroids of the Delaunay triangles and visa-versa. Such a condition is effectively an implicit constraint on the *regularity* of the grid, with an increase in the *centroidal*-ness of a grid associated with improvements in the shape of its Delaunay triangles and Voronoi polygons. Centroidal grids typically lead to high-quality numerical discretisations, with grids containing near-perfect element configurations achieving *optimal* convergence rates. The unstructured C-grid scheme outlined previously is known to achieve fully second-order accurate convergence when applied to centroidal Voronoi grids (**?**).

While a number of unstructured grid-generation algorithms currently exist, as outlined in Section 1, I am not aware of any that are successful in generating the very high-quality *locally-orthogonal*, *well-centred* and *centroidal* grids required by unstructured C-grid type general circulation models. As a result, throughout the remainder of this paper, the development of methods for the generation of such staggered Delaunay/Voronoi tessellations is pursued in detail.

**2.1  Grid-quality metrics**

[revised manuscript text omitted]

$$a(f_i) = \frac{4\sqrt{3}}{3} \frac{A_f}{\|\mathbf{e}_{\mathrm{rms}}\|^2},$$

$$h_r(e_j) = \frac{\|\mathbf{e}_j\|}{\bar{h}(\mathbf{x}_m)},$$

In this study, a high-quality triangular surface mesh is generated on the spheroidal reference surface (4) using a *Frontal-Delaunay* variant of the conventional restricted Delaunay-refinement algorithm (**?????**). This technique is described by the
* * *
**Algorithm 1** Restricted Frontal-Delaunay Refinement
* * *
1: **function** DELFRONT($\Sigma, \bar{\rho}, \bar{h}(\mathbf{x}), \text{Del}\,|_\Sigma(X)$)

2:    Place 12 vertices on the spheroidal surface $\Sigma$ to form a regular icosahedron. Initialise the full Delaunay complex Del($X$) and the restricted surface triangulation Del$\,|_\Sigma(X)$.

3:    **while** ($\exists$ BADTRIANGLE(Del$\,|_\Sigma(X)$)) **do**

4:        Find the worst bad *frontal* triangle in the surface triangulation $f_i \leftarrow$ Del$\,|_\Sigma(X)$.

5:        Call $\mathbf{x}_i \leftarrow$ OFFCENTRE($f_i$) to form the new off-centre vertex $\mathbf{x}_i$, designed to eliminate the bad triangle $f_i$.

6:        Push Del($X$) $\leftarrow$ $\mathbf{x}_i$ and update the restricted surface triangulation Del$\,|_\Sigma(X)$.

7:    **end while**

8:    **return** surface triangulation object Del$\,|_\Sigma(X)$.

9: **end function**

1: **function** BADTRIANGLE($f$)

    if ($h(f) > \bar{h}(\mathbf{x}_f)$)    **return** TRUE

2:    if ($\rho_2(f) > \bar{\rho}$)    **return** TRUE

    else **return** FALSE

3: **end function**

1: **function** OFFCENTRE($f, \mathbf{x}$)

2:    Form the restricted triangle circumcentre $\mathbf{x}_c \leftarrow$ Vor$\,|_f(X) \cap \Sigma$ and the associated surface ball B = SDB($f$).

3:    Find the minimum length edge $e$ in the triangle $f$. Edge $e$ is the *frontal* segment.

4:    Place the *size-optimal* point $\mathbf{x}_h$ along the arc Vor$\,|_e(X) \cap \Sigma$ to satisfy local spacing constraints $\bar{h}(\mathbf{x})$.

5:    Place the *shape-optimal* point $\mathbf{x}_\theta$ along the arc Vor$\,|_e(X) \cap \Sigma$ to satisfy minimum angle constraints $\rho(f) \leq \bar{\rho}$.

6:    Compute the distances $d_h = \text{dist}(\mathbf{x}_e, \mathbf{x}_h)$ and $d_\theta \leftarrow \text{dist}(\mathbf{x}_e, \mathbf{x}_\theta)$, where $\mathbf{x}_e$ is the midpoint of the short edge $e$.

7:    Set $\mathbf{x}$ to the point $\{\mathbf{x}_h, \mathbf{x}_\theta\}$ of minimum distance $d_j$, such that $d_j \geq \frac{1}{2}\|\mathbf{e}\|$. Fall-back to $\mathbf{x} \leftarrow \mathbf{x}_c$ if $\{\mathbf{x}_h, \mathbf{x}_\theta\}$ do not satisfy constraints.

8:    **return** off-centre vertex $\mathbf{x}$.

9: **end function**
* * *
**3.5**

author in detail in **??** and differs from standard Delaunay-refinement approaches in terms of  the strategies used for the placement of new Steiner vertices. Specifically, the Frontal-Delaunay  algorithm employs a generalisation of various *off-centre* type point-placement techniques (**??**), designed to position vertices  such that that element-quality and mesh-size constraints are satisfied in a *locally-optimal* fashion. Previous studies have shown that such an approach typically

5  leads to substantial improvements in mean element-quality and mesh smoothness.

    While a variety of grid-generation algorithms for the surface meshing problem have been described previously (e.g. **????**), the restricted Frontal-Delaunay method  presented here has been found to offer a unique combination of characteristics — combining the smooth, high quality grid-generation capabilities of an advancing-front approach, with the

10 theoretical robustness  and provable guarantees associated with conventional Delaunay-refinement techniques. Specifically, the Frontal-Delaunay approach presented here is known to generate grids with very high mean element quality, bounded minimum

and maximum angles, tight conformance to grid-spacing constraints, and provable guarantees on topological consistency and convergence. A full description of the algorithm, including detailed discussions of its theoretical foundations and proofs of worst-case grid-quality bounds, can be found in **??**.

Given a user-defined mesh-spacing function $\bar{h}(\mathbf{x})$ and an upper-bound on the element *radius-edge* ratios $\bar{\rho} \geq 1$, the Frontal-Delaunay algorithm proceeds to sample the spheroidal surface $\Sigma$ by refining any surface triangle that violates either the mesh-spacing or element-quality constraints.  The full algorithm is described in Algorithm 1, and detailed snap-shots of the refinement process are shown in Figure 5. The surface is seeded with a set of twelve vertices to form a standard icosahedron. Refinement then proceeds to insert a new Steiner vertex at the  *off-centre* refinement point associated with  each given element to be *eliminated*. Refinement continues until all constraints are satisfied. The refinement process is priority scheduled, with triangles $f_i \in \mathrm{Del}|_\Sigma(X)$ ordered according to their radius-edge ratios $\rho(f_i)$. This ordering ensures that the element with the *worst* ratio is refined at each iteration. Additionally, triangles are subject to a *frontal* filtering — requiring that a low-quality element be adjacent to a *converged* triangle before being considered for refinement. This logic helps the algorithm mimic the behaviour of *advancing-front* type algorithms, with new vertices and elements expanding from initial seeds. Upon termination, the resulting surface triangulation is guaranteed to contain nicely shaped elements, satisfying both the radius-edge constraints $\rho(f_i) \leq \bar{\rho}$ and mesh-spacing bounds $h(f_i) \leq \bar{h}(\mathbf{x}_f)$ for all surface triangles $f_i \in \mathrm{Del}|_\Sigma(X)$ in the mesh. Setting $\bar{\rho} = 1$ guarantees that element angles are bounded, such that $30° \leq \theta_f \leq 120°$, ensuring that the grid does not contain any highly distorted elements.

**3.5 Off-centre point-placement**

The performance of the Frontal-Delaunay  algorithm hinges on the use of *off-centre* refinement rules —  locally-optimal point-placement strategies designed to create very high-quality vertex distributions. A set of candidate off-centre points are considered at each new vertex insertion. Type I vertices, $\mathbf{x}_c$, are equivalent to conventional element circumcentres (positioned at the centre of the associated surface balls), and are used to preserve global convergence. Type II vertices, $\mathbf{x}_h$, are so-called *size-optimal* points, and are designed to satisfy grid-spacing constraints in a locally optimal fashion. Type III vertices, $\mathbf{x}_\theta$, are so-called *shape-optimal* points, and are designed to ensure minimum-angle bounds are satisfied in a worst-first manner. The Type II and Type III strategies employed here can be seen as a generalisation of the two-dimensional off-centre techniques presented by **?** and **?**, respectively. See Figure 6 for additional detail.

Given a low-quality triangle $f_i \in \mathrm{Del}|_\Sigma(X)$, the Type II and Type III vertices $\mathbf{x}_h$ and $\mathbf{x}_\theta$ are positioned along the intersection of an adjacent segment of the Voronoi complex and the spheroidal surface. This intersection $\mathrm{Vor}|_e(X) \cap \Sigma$, defines a *frontal-curve* inscribed on $\Sigma$ —  a *locally-optimal* geodesic segment on which to insert new vertices. Here, $\mathrm{Vor}|_e(X)$ is the polygonal face of the Voronoi complex associated with the short edge $e_0 \in f_i$. In the context of conventional advancing-front type methods, the edge $e_0$ would denote the current *frontal-segment* about which vertex insertion occurs.

The points $\mathbf{x}_h$ and $\mathbf{x}_\theta$ are positioned to form new candidate triangles about the frontal edge $e_0$, such that local constraints are satisfied optimally. The size-optimal point $\mathbf{x}_h$ is positioned to adhere to local grid-spacing constraints $\bar{h}(\mathbf{x})$, with the altitude

[Figure]

**Figure 5.** Progress of the Frontal-Delaunay refinement algorithm, clockwise from top-left. New vertices are inserted incrementally until all constraints on element-shape and edge-length are satisfied globally. In contrast with standard advancing-front type grid-generation techniques, new vertices are inserted according to a *greedy* priority-schedule. Grid-generation proceeds along a pseudo *space-filling* trajectory as a result.

of the new triangle candidate calculated to define correctly sized edges, such that $\|\mathbf{e}_1\| \leq \bar{h}(\mathbf{m}_1)$ and $\|\mathbf{e}_2\| \leq \bar{h}(\mathbf{m}_2)$, where the $\mathbf{m}_i$'s are the edge midpoints. These constraints can be solved for an associated altitude

$$a_h = \min\left(a_h^*, \sqrt{3}/2\,\bar{h}\right), \quad \text{where:} \quad a_h^* = \sqrt{\bar{h}^2 - \tfrac{1}{2}\|\mathbf{e}_0\|^2}, \quad \bar{h} = \tfrac{1}{2}\left(\bar{h}(\mathbf{m}_1) + \bar{h}(\mathbf{m}_2)\right) \tag{6}$$

Similarly, the shape-optimal point is positioned to adhere to minimum angle constraints, sliding the point $\mathbf{x}_\theta$ along the inscribed curve $\mathrm{Vor}|_\varepsilon(X) \cap \Sigma$ such that the new triangle candidate satisfies $\rho \leq \bar{\rho}$. Setting $\rho = \bar{\rho}$ leads to a solution for the associated shape-optimal altitude

$$a_\theta = \frac{\|\mathbf{e}_0\|}{2\tan\left(\tfrac{1}{2}\bar{\theta}\right)}, \quad \text{where:} \quad \bar{\theta} = \arcsin\left(\frac{1}{2\bar{\rho}}\right) \tag{7}$$

The position of the points $\mathbf{x}_h$ and $\mathbf{x}_\theta$ is calculated by computing the intersection of balls of radius $a_h$ and $a_\theta$, centred at the midpoint of the frontal edge $e_0$ and the *frontal* curve $\mathrm{Vor}|_\varepsilon(X) \cap \Sigma$. This approach ensures that new vertices are positioned by advancing a specified distance along the surface $\Sigma$ in the *frontal* direction. For non-uniform $\bar{h}(\mathbf{x})$, expressions for the position of the point $\mathbf{x}_h$ are non-linear, with the altitude $a_h$ depending on an evaluation of the mesh-size function at the edge midpoints $\bar{h}(\mathbf{m}_i)$ and visa-versa. In practice, since $\bar{h}(\mathbf{x})$ is guaranteed to be sufficiently smooth, a simple iterative predictor-corrector procedure is sufficient to solve these expressions approximately.

Given the set of candidate vertices $\{\mathbf{x}_c, \mathbf{x}_h, \mathbf{x}_\theta\}$, the position of the refinement point $\mathbf{x}$ for the triangle $f_i$ is selected. A *worst-first* strategy is adopted, choosing the point that satisfies local constraints in a greedy fashion. Specifically, the closest point lying on the adjacent Voronoi segment $\mathrm{Vor}|_\varepsilon(X)$ and outside the neighbourhood of the frontal edge $e_0$ is selected, with

$$\mathbf{x} = \begin{cases} \mathbf{x}_j, & \text{if } \left(d_j \leq d_c \text{ and } d_j \geq \tfrac{1}{2}\|\mathbf{e}_0\|\right) \\ \mathbf{x}_c, & \text{otherwise} \end{cases}, \quad \text{where:} \quad j = \operatorname{argmin}(d_h, d_\theta). \tag{8}$$

Here, $d_j = \mathrm{dist}(\mathbf{x}_e, \mathbf{x}_j)$ are distances from the midpoint of the frontal edge $e_0$ to the size- and shape-optimal points $\mathbf{x}_h$ and $\mathbf{x}_\theta$. The cascading selection criteria (8) seeks a balance between local optimality and global convergence, smoothly degenerating to a conventional circumcentre-based Delaunay-refinement strategy in limiting cases, while using locally optimal points where possible. Specifically, these constraints guarantee that refinement points lie within a local *safe* region on the Voronoi complex — being positioned on an adjacent Voronoi segment and bound between the circumcentre of the element itself and the diametric ball of the associated frontal edge. These constraints ensure that new points are never positioned too close to an existing vertex, leading to provable guarantees on the performance of the algorithm. See previous work by the author (**??**) for additional detail.

**3.6   Additional remarks**

As a *restricted* Delaunay-refinement approach, a full three-dimensional Delaunay tetrahedralisation $\mathrm{Del}(X)$ is incrementally maintained throughout the surface meshing phase, where $X \in \mathbb{R}^3$ is the set of vertices positioned on the surface of

[Figure]

**Figure 6.** A two-dimensional representation of the off-centre refinement strategies utilised for a given low-quality triangle $f_i$ (shaded), illustrating (a) the *frontal* segment of the Voronoi diagram associated with the short edge $e_0 \in f_i$, (b) placement of the size-optimal vertex $x_h$ such that local size constraints $\bar{h}(x_f)$ are enforced, and (c) placement of a shape-optimal vertex $x_\theta$ such that the required radius-edge ratio $\bar{\rho}$ is satisfied.

the spheroidal geometry. The set of restricted surface triangles $\text{Del}|_\Sigma(X)$ that conform to the underlying spheroidal geometry are expressed as a subset of the tetrahedral faces, such that $\text{Del}|_\Sigma(X) \subseteq \text{Del}(X)$. In an effort to minimise the expense associated with maintaining the full-dimensional topological tessellation, an additional *scaffolding* vertex $x_s$ is initially inserted at the centre of the spheroid. This has the effect of simplifying the resulting topological structure of the interior mesh, with the resulting tetrahedral elements forming a simple *wheel-like* configuration, in which they emanate radially outward from the central scaffolding vertex $x_s$.

**3.7**

~~Construction of the staggered surface Voronoi control-volumes, illustrating: (left) locally-orthogonal Voronoi/Delaunay staggering, and (right) an associated unstructured C-grid type numerical discretisation scheme, as per the MPAS model. The formulation is a combination of conservative *cell-centred* tracer quantities $\bar{\psi}_i$, *edge-centred* normal velocity components $(u \cdot \hat{n})_j$, and auxiliary *vertex-centred* vorticity variables $\xi_k$.~~

~~Typically, numerical formulations employed for large-scale atmospheric and/or oceanic general circulation modelling are based on a *staggered* grid configuration, with quantities such as fluid pressure, geopotential, and density discretised using a primary control-volume, and the fluid velocity field and vorticity distribution represented on a second, spatially distinct grid-cell. In the context of standard structured grid types, various staggered arrangements are described by the well-known Arakawa schemes (?)~~ See Figure 4 for additional detail.

~~The development of general circulation models based on unstructured grid types is an emerging area of research, and, as a result, a variety of numerical formulations are currently under investigation. In this study, the development of *locally-orthogonal* grids appropriate for staggered unstructured numerical schemes are pursued, as these methods are thought to represent the most logical extension of the conventional structured Arakawa type techniques to the unstructured setting.~~

Noting that the As per Figure 5, an interesting characteristic of the Frontal-Delaunay refinement algorithm described previously is guaranteed to construct triangulations that respect the Delaunay criterion, a natural staggered unstructured grid can be constructed based on the associated Voronoi diagram.

Consisting of a set of (convex) polygonal grid-cells centred on each vertex in the underlying triangulation, algorithm described here relates to the surface Voronoi diagram $\mathrm{Vor}|_\Sigma(X)$ obeys a number of local orthogonality constraints. Specifically, grid-cell edges in the Voronoi tessellation are guaranteed to be perpendicular to their associated edges in the underlying Delaunay triangulation, passing through the Delaunay-edge midpoints. Additionally, *refinement-trajectory* that is followed when inserting new vertices and triangles. Unlike standard Delaunay-refinement or advancing-front type methods, it can be seen that the algorithm adopts a *space-filling curve* type pattern, covering the surface in a fractal-like configuration, before recursively filling in the gaps. Note that no explicit space-filling curve constraint has been implemented here — this behaviour is simply an emergent property of the algorithm itself, due to interactions between the greedy priority schedule, the ease of perfectly *regular* and *centroidal* tessellations, the Delaunay-edges are guaranteed to pass through the midpoints of their associated Voronoi duals. Voronoi grid-cells are formed as the convex-hull of the incident element surface-ball centres $B(\mathbf{c}_i, r_i)$ associated with the set of surface triangles adjacent to each vertex. Example Voronoi/Delaunay type grid staggering is illustrated in Figure 2.

While detailed comparisons of particular numerical discretisation schemes lie outside the scope of the current study, brief comments regarding the benefits of locally-orthogonal grid-staggering arrangements are made. Pursuing an unstructured variant of the widely-used Arakawa C-grid, the placement of fluid pressure, geopotential and density degrees-of-freedom within the primary Voronoi control-volumes, and orthogonal velocity vectors on Delaunay-edges achieves a similar configuration. Such an arrangement facilitates construction of a standard conservative finite-volume type scheme for the transport of fluid properties and a *mimetic* class (**??**) finite-difference formulation for velocity components. Additionally, exploiting alignment with Delaunay-edges, a conservative evaluation of the fluid vorticity can be made on the staggered Delaunay triangles. This type of *unstructured* C-grid staggering is employed in the Model for Prediction Across Scales (MPAS), for both atmospheric and oceanic modelling (**???**). See Figure 2 for additional details frontal filtering and off-centre point-placement strategies. In practice, it has been found that this space-filling type behaviour leads to the construction of very high-quality triangulations, typically exceeding the performance of standard advancing-front type schemes.

**4    Hill-climbing mesh optimisation**

While the staggered Voronoi/spheroidal Delaunay grids generated using the Frontal-Delaunay refinement algorithm described in Section 3 are guaranteed to be of very high-quality, producing triangulations with angles bounded between $\theta_f = 30$ and $\theta_f = 120$ degrees, these tessellations can often be further improved through subsequent *mesh-optimisation* operations. Such a procedure Recalling that the construction of the *well-centred* grids appropriate for unstructured C-grid schemes *require* that maximum angles be bounded below $\theta_f = 90°$, the application of such optimisation procedures can in fact be seen as a *necessary* component of the grid-generation work-flow for such models.

[revised manuscript text omitted]

**Algorithm 2** Spring and gradient-based node-smoothing

1: **function** NODESMOOTH($\mathbf{x}$, Del $|_\Sigma(X)$, $\bar{Q}$)

2:   **if** $(\min(Q(\mathbf{x})) \geq \bar{Q})$ **then**

3:     Call $\{\hat{\mathbf{v}}, \Delta\} \leftarrow$ SPRINGVEC($\mathbf{x}$, Del $|_\Sigma(X)$) to form the *spring-based* search vector and step-length.

4:   **else**

5:     Call $\{\hat{\mathbf{v}}, \Delta\} \leftarrow$ ASCENTVEC($\mathbf{x}$, Del $|_\Sigma(X)$) to form the *gradient-based* search vector and step-length.

6:   **end if**

7:   **while** $(i \leq M)$ **do**

8:     Set $\mathbf{x}' \leftarrow$ proj $|_\Sigma(\mathbf{x} + \Delta^i \hat{\mathbf{v}})$ to move the vertex $\mathbf{x}$ along the search vector $\hat{\mathbf{v}}$ and project onto the surface $\Sigma$.

9:     **if** $(\text{BETTER}(Q(\mathbf{x}'), Q(\mathbf{x})))$ **then**

10:       Set $\mathbf{x} \leftarrow \mathbf{x}'$ and **break**.

11:     **end if**

12:     Step-length bisection $\Delta^{i+1} \leftarrow \frac{1}{2}\Delta^i$.

13:   **end while**

14: **end function**

1: **function** SPRINGVEC($\mathbf{x}$, Del $|_\Sigma(X)$, $\hat{\mathbf{v}}$, $\Delta$)

2:   Scan edges adj. to $\mathbf{x}$, calc. $\mathbf{x}' \leftarrow \frac{\sum w_e(\mathbf{x} + \Delta_e \mathbf{e})}{\sum w_e}$, where $w_e \leftarrow \Delta_e^2$ and $\Delta_e \leftarrow \frac{h(\mathbf{x}_e) - \|\mathbf{e}\|}{\|\mathbf{e}\|}$.

3:   Set $\mathbf{v} \leftarrow \mathbf{x}' - \mathbf{x}$, $\hat{\mathbf{v}} \leftarrow \frac{\mathbf{v}}{\|\mathbf{v}\|}$ and $\Delta \leftarrow \|\mathbf{v}\|$.

4:   **return** search vector $\hat{\mathbf{v}}$ and step-size $\Delta$.

5: **end function**

1: **function** ASCENTVEC($\mathbf{x}$, Del $|_\Sigma(X)$, $\hat{\mathbf{v}}$, $\Delta$)

2:   Scan faces adj. to $\mathbf{x}$ and return the worst adj. element $f \leftarrow \text{argmin}(Q_f(\mathbf{x}))$.

3:   Compute the local *gradient-ascent* search vector $\mathbf{v} \leftarrow \frac{\partial}{\partial \mathbf{x}}(Q_f(\mathbf{x}))$.

4:   Solve $Q_f(\mathbf{x}) + \Delta \hat{\mathbf{v}} \cdot \frac{\partial}{\partial \mathbf{x}}(Q_f(\mathbf{x})) \leq \tilde{Q}_j(\mathbf{x})$ for the initial step-size $\Delta$, where $\tilde{Q}_j(\mathbf{x})$ is the quality of the *second-worst* triangle adj. to $\mathbf{x}$.

5:   **return** search vector $\hat{\mathbf{v}}$ and step-size $\Delta$.

6: **end function**

1: **function** BETTER($Q'$, $Q$)

2:   Sort the mesh-quality vectors $Q'$ and $Q$.

3:   **return** FALSE if any $Q_i' < Q_i$, else TRUE.

4: **end function**

[revised manuscript text omitted]

**Algorithm 3** Hill-climbing grid optimisation

1: **function** OPTIMISE($\Sigma, \bar{h}(\mathbf{x}), \mathrm{Del}|_\Sigma(X)$)
2:     **for** ($i_{\mathrm{outer}} \leq N_{\mathrm{outer}}$) **do**
3:         **for** ($i_{\mathrm{inner}} \leq N_{\mathrm{inner}}$) **do**
4:             **for** ($\forall$ nodes $j \in X$) **do**
5:                 Call NODESMOOTH($\mathbf{x}_j, \mathrm{Del}|_\Sigma(X)$) to improve geometric distribution of triangle vertices.
6:             **end for**
7:         **end for**
8:         Call MERGENODE($\mathrm{Del}|_\Sigma(X)$) to collapse any *short* edges.
9:         Call SPLITEDGE($\mathrm{Del}|_\Sigma(X)$) to refine any *long* edges.
10:       Call EDGEFLIPS($\mathrm{Del}|_\Sigma(X)$) to restore grid Delaunay-*ness* (i.e. local-orthogonality).
11:     **end for**
12:     **return** surface triangulation object $\mathrm{Del}|_\Sigma(X)$.
13: **end function**

1: **function** EDGEFLIPS($X, \mathrm{Del}|_\Sigma(X)$)
2:     **for** ($\forall$ edges $e \in \mathrm{Del}|_\Sigma(X)$) **do**
3:         Given the adj. triangle-pair $\mathscr{C} \leftarrow \{f_i, f_j\}$, *flip* the common diagonal to re-triangulate the cavity $\mathscr{C}'$.
4:         **if** BETTER($Q(\mathscr{C}'), Q(\mathscr{C})$), $\mathrm{Del}|_\Sigma(X) \leftarrow \mathscr{C}'$.
5:     **end for**
6: **end function**

1: **function** MERGENODE($X, \mathrm{Del}|_\Sigma(X)$)
2:     **for** ($\forall$ edges $e \in \mathrm{Del}|_\Sigma(X)$) **do**
3:         Form a local *merge* point $\mathbf{x}'$ about the edge $e$. Given $\mathbf{x}_i, \mathbf{x}_j \in e$, set $\mathbf{x}' \leftarrow \mathrm{proj}|_\Sigma(\frac{1}{|\mathscr{C}|} \sum \mathbf{c}_f)$, where $\mathscr{C}$ is the set of triangles adj. to $\mathbf{x}_i, \mathbf{x}_j$ and $\mathbf{c}_f$ are the associated circumcentres.
4:         Merge the vertices $\mathbf{x}_i, \mathbf{x}_j \leftarrow \mathbf{x}'$, and re-triangulate the cavity $\mathscr{C}'$ as a result.
5:         **if** BETTER($Q(\mathscr{C}'), Q(\mathscr{C})$), $\mathrm{Del}|_\Sigma(X) \leftarrow \mathscr{C}'$.
6:     **end for**
7:     **return** updated $X$ and $\mathrm{Del}|_\Sigma(X)$.
8: **end function**

1: **function** SPLITEDGE($X, \mathrm{Del}|_\Sigma(X)$)
2:     **for** ($\forall$ edges $e \in \mathrm{Del}|_\Sigma(X)$) **do**
3:         Form a local *refinement* point $\mathbf{x}'$ about the edge $e$. Given the two adj. triangles, $\mathscr{C} \leftarrow \{f_i, f_j\}$, set $\mathbf{x}'$ equal to the circumcentre of the lower quality element.
4:         Append the new vertex $X \leftarrow \mathbf{x}'$, and re-triangulate the cavity $\mathscr{C}'$ as a result.
5:         **if** BETTER($Q(\mathscr{C}'), Q(\mathscr{C})$), $\mathrm{Del}|_\Sigma(X) \leftarrow \mathscr{C}'$.
6:     **end for**
7:     **return** updated $X$ and $\mathrm{Del}|_\Sigma(X)$.
8: **end function**

[revised manuscript text omitted]

**5.5 Computational performance**

In addition to the generation of very high-quality grids, the new JIGSAW-GEO algorithm also imposes a relatively moderate computational burden, producing large-scale, multi-resolution grids in a matter of minutes using standard

15  desktop-based computing infrastructure. Specifically, grid-generation for the UNIFORM-SPHERE, REGIONAL-ATLANTIC and SOUTHERN-OCEAN test-cases required 12 seconds, $1\frac{1}{2}$ minutes and 10 minutes of computation time, respectively, running on a single core of an Intel i7 processor. In all cases, grid-optimisation was found to be approximately four times as expensive as the initial Frontal-Delaunay refinement. Compared to the existing iterative MPI-SCVT algorithm (?), commonly used to generate grids for the MPAS framework, these results represent a significant increase in productivity, with the

20  MPI-SCVT algorithm often requiring days, or even weeks of distributed computing time.

Additionally, practical experience with the MPI-SCVT algorithm has shown that it cannot always be relied upon to generate an appropriate grid, irrespective of the amount of computational time allowed for convergence to be reached. While always generating a *locally-orthogonal* and *centroidal* Voronoi tessellation with very high mean grid-quality, the MPI-SCVT algorithm does not provide bounds on worst-case grid-quality. In practice, multi-resolution grids generated using the MPI-SCVT

25  algorithm are often observed to contain a minority of obtuse triangles that violate the *well-centred* constraint, and, due to the nature of the numerical formulation, such grids are inappropriate for use in an unstructured C-grid model such as the MPAS framework. Grid generation for such models often requires a degree of user-driven *trial-and-error* as a result, making grid-generation a somewhat arduous task for model-users. Initial experiments conducted using the JIGSAW-GEO algorithm have shown it to be a useful alternative, reliably generating valid well-centred multi-resolution grids for the MPAS ocean and

30  land-ice frameworks given a wide range of user-defined constraints and configuration settings.

**6 Conclusions & Future Work**

A new algorithm for the generation of  multi-resolution staggered unstructured grids for large-scale general circulation modelling on the sphere has been described. Using a combination of Frontal-Delaunay refinement and hill-climbing type optimisation techniques, it has been shown that very high-quality  *locally-orthogonal*, *centroidal* and *well-centred* spheroidal grids appropriate for unstructured C-grid type general circulation models can be generated. The performance of this new approach has been verified using a number of multi-scale global benchmarks, including difficult problems incorporating highly non-uniform mesh-spacing constraints.

This new algorithm is available as part of the JIGSAW meshing package, providing a simple and easy-to-use tool for the oceanic and atmospheric modelling communities. A number of global-scale benchmark problems have been analysed,  examining the performance of the new approach. The Frontal-Delaunay refinement algorithm has been shown to generate *guaranteed-quality* spheroidal Delaunay triangulations — satisfying worst-case bounds on element-wise angles and exhibiting smooth grading characteristics. This algorithm has been shown to produce very high-quality multi-resolution triangulations, with a majority of elements exhibiting near-perfect conformance to element-shape and grid-spacing based constraints.

The use of a coupled geometrical and topological hill-climbing type optimisation procedure was shown to further improve  grid-quality statistics, especially for the lowest quality elements in each mesh. It was demonstrated that these optimisation techniques allow grid-quality to be improved to the extent that fully *well-centred* mesh configurations  can be achieved, with all angles in the surface triangulation bounded below $90°$. For the three global test-cases presented here, enclosed-angles were bounded above $\theta_f \geq 40°$ and below $\theta_f \leq 80°$.

The construction of locally-orthogonal staggered polygonal grids appropriate for a range of contemporary unstructured C-grid type general circulation models  was discussed in detail, with a focus on the generation of multi-resolution grids for the MPAS framework. The availability of this new algorithm is expected to significantly reduce the grid-generation burden for MPAS model-users. Future work will focus on a generalisation of the algorithm and improvements to its efficiency, including: (i) support for inscribed geometrical constraints, such as coastlines, (ii) the use of multi-threaded programming patterns to improve computational performance, and (iii) further enhancements to the mesh optimisation procedures, with a focus on improving the *well-centredness* of the resulting staggered grids. The investigation of *solution-adaptive* multi-scale representations, in which grid-resolution is adapted to spatial variability in model state (**?**), is also an obvious direction for future investigation.

*Acknowledgements.* This work was carried out at the NASA Goddard Institute for Space Studies, the Massachusetts Institute of Technology, and the University of Sydney with the support of a NASA–MIT cooperative agreement and an Australian Postgraduate Award. The author wishes to thank  Todd Ringler, Luke Van Roekel, Mark Petersen, Matthew Hoffman and

Phillip Wolfram for their assistance on grid-generation for the MPAS-O environment. John Marshall provided feedback on an earlier version of the manuscript.

The author also wishes to thank the anonymous reviewers for their helpful comments and feedback.

**7 Code availability**

5    The JIGSAW-GEO grid-generator can be found online at github.com/dengwirda/jigsaw-geo-matlab

**Appendix A: Spheroidal Predicates**

Recalling the methodology described in Section 3, computation of the *restricted* Delaunay surface tessellation $\text{Del}|_\Sigma(X)$ requires the evaluation of a single geometric predicate. Given a spheroidal surface $\Sigma$, the task is to compute intersections between edges in the Voronoi tessellation $\text{Vor}(X)$ and the input features $\Sigma$.

10 ## A1    Restricted surface triangles

Restricted surface triangles $f_i \in \text{Del}|_\Sigma(X)$ are defined as those associated with an *intersecting* Voronoi edge $v_e \in \text{Vor}(X)$, where $v_e \cap \Sigma \neq \emptyset$. These triangles provide a good piecewise linear approximation to the surface $\Sigma$. For a given triangle $f_i$, the associated Voronoi edge $v_e$ is defined as the line-segment joining the two *circumcentres* $\mathbf{c}_i$ and $\mathbf{c}_j$ associated with the pair of tetrahedrons that share the face $f_i$. The task then is to find intersections between the line-segments $v_e$ and the surface $\Sigma$.

15    Let $\mathbf{p}$ be a point on a given Voronoi edge-segment $v_e$

$$\mathbf{p} = \bar{\mathbf{c}} + t\Delta, \quad -1 \leq t \leq +1, \text{where:} \quad \text{where:} \quad \bar{\mathbf{c}} = \tfrac{1}{2}(\mathbf{c_i} + \mathbf{c_j}), \quad \Delta = \tfrac{1}{2}(\mathbf{c_j} - \mathbf{c_i}). \tag{A1}$$

Substituting (A1) into the expression for the spheroidal surface (4), the existence of real, bounded solutions, such that $-1 \leq t \leq +1$, indicates a non-trival intersection $v_e \cap \Sigma \neq \emptyset$. Specifically, expanding and rearranging after substitution

$$\sum_{i=1}^{3} \left( \frac{\bar{\mathbf{c}}_i + t\Delta_i}{r_i} \right)^2 = 1, \quad \sum_{i=1}^{3} \frac{\bar{\mathbf{c}}_i^2 + 2t\bar{\mathbf{c}}_i\Delta_i + t^2\Delta_i^2}{r_i^2} = 1, \quad \sum_{i=1}^{3} \left( \frac{\Delta_i^2}{r_i^2} \right) t^2 + \left( \frac{2\bar{\mathbf{c}}_i\Delta_i}{r_i^2} \right) t + \left( \frac{\bar{\mathbf{c}}_i^2}{r_i^2} - 1 \right) = 0, \tag{A2}$$

20    which is simply a quadratic expression for the parameter $t$ and can be solved using the standard approach. Given a real solution $-1 \leq t_\Sigma \leq +1$ the corresponding point of intersection $\mathbf{p}_\Sigma$ can be found using (A1).

Geosci. Model Dev. Discuss.,
doi:10.5194/gmd-2016-296-AC1, 2017

[Figure]

Thanks for your comments and recommendations. Following your suggestions, I propose to update the title of the paper to:

JIGSAW-GEO (1.0): Locally-orthogonal staggered unstructured grid-generation for general circulation modelling on the sphere

Please let me know if you have further suggestions or comments regarding the submission.

[Figure]

[Figure]

Kind regards,

Darren Engwirda
* * *
Geosci. Model Dev. Discuss.,
doi:10.5194/gmd-2016-296-AC2, 2017

[Figure]
Thank you for your comments and recommendations regarding the draft manuscript. I propose to incorporate many of your suggested changes 'as-is' in the revised submission. In some cases, as detailed below, I do not fully agree with the comments made, and have attempted to reply in detail to some of the points raised. In all cases I found the suggestions helpful, and look forward to amending the revised manuscript in response to your review.

Overall, I agree with the suggestion that the revised paper should better articulate the

impact of the techniques described in the present work, and better differentiate them from existing approaches. I include a detailed set of proposed changes below.

Reviewer comments are presented in italics, my responses are included in plain-text.

*The current paper deals with high quality surface triangulation applied to general circulation modelling. The paper bypasses a parametric representation of an arbitrary surface by limiting the surface definition to an ellipsoid representing the earth. The main algorithm relies on a coupled Frontal-Delaunay approach. Various examples are provided to illustrate the method.*

*Overall, the paper is clear and there is obviously a lot of work in it. However, my main critic is that there is not much new brought by the paper, as opposed to what is claimed in the conclusion, except for the fact of applying it to a general circulation modelling. The curvature of the Earth is almost constant so technically it is not difficult to surface mesh it. The paper introduces a lot of concepts such as restricted Delaunay, Frontal Delaunay, Hill-climbing, Gradient based smoothing, etc, which are very classic and well established unstructured mesh techniques. At least, it should be clearly stated what is new.*

*1. The claim that Voronoi edges are always perpendicular to mesh edges is wrong. It is only valid for an acute triangulation, which is not true in general, particularly because of the boundaries. This is a property that has been pursued by the electromagnetic solvers for a long time for the same reason but only partially reached. This is only briefly mentioned in Section 4.*

With respect, I don't believe the reviewer to be correct here. Voronoi edges are indeed (always) perpendicular to their associated Delaunay edges, even when the triangulation is non-acute. This local orthogonality is a fundamental aspect of the Voronoi-Delaunay geometric duality, and lies at the heart of many unstructured numerical formulations that are based on a Voronoi-Delaunay grid staggering, including the discretisation scheme employed in the MPAS framework referenced in the current paper.

[Figure]

What the reviewer may be referring to is the notion of 'well-centred' Voronoi-Delaunay grid staggering, which is a property only realised for 'acute' triangulations (all angles less than 90 degrees). The properties and benefits of 'well-centred' Voronoi-Delaunay staggered grids are discussed in the present paper in Section 4.

Voronoi vertices (the circumcentres of Delaunay triangles) only lie 'within' their associated Delaunay triangles when the triangulation is acute. As a result, Delaunay triangles containing obtuse angles induce an undesirable grid-staggering, with the Voronoi edges associated with these large angles failing to intersect their paired Delaunay edges.

In the context of unstructured general circulation models (i.e. the MPAS model) building such 'well-centred' Voronoi-Delaunay grids is (a) important (the MPAS framework, for instance, requires such constraints to be satisfied to ensure a correct evaluation of the vertex-centred vorticity distribution), and (b) non-trivial (a majority of conventional unstructured meshing algorithms do not produce such grids).

As such, the development of algorithms designed to produce high-quality well-centred grids is, in my view, a new and useful result. I am not aware of other publicly available software for the oceanic/atmospheric modelling communities that offers similar functionality.

In the current work, it's shown that a combination of Frontal-Delaunay refinement and hill-climbing optimisation is an effective strategy — able to produce very high-quality well-centred Voronoi-Delaunay grids even when complex, highly non-uniform grid sizing constraints are imposed. I believe this to be a new result of benefit to the unstructured oceanic/atmospheric modelling communities. Public availability of the associated JIGSAW-GEO grid-generator is also thought to be a further benefit to the community.

Noting the reviewers concerns, and subsequent comments below, I propose to re-work the paper to make the arguments above more clearly, and earlier in the manuscript. I propose explicitly defining the notion of 'well-centred' grids in Section 2, and to amend

the abstract/introductory sections accordingly. I propose to include a new figure illustrating the distinction between 'well-centred' and 'non-well-centred' Voronoi-Delaunay grids.

*2. Line 28 (page 4). It is not clear at all in general that maximisation of the minimum angle is beneficial. Add references.*

Agreed. I will include additional references/explanation in the revised manuscript.

*3. The pictures do not clearly show the mesh transitions for size variations in details.*

Agreed. I will include better images of transitional mesh regions.

*4. The abstract mentions a-priori guaranteed quality bounds while nothing is proved. Empirical studies show good results but no bounds are provided.*

Such bounds are rigorously established in the companion paper (Engwirda and Ivers, 2016) that describes the Frontal-Delaunay refinement algorithm in full, with this point mentioned in Section 2.5 (pages 7-8) where references to the companion paper are given and results of the proofs summarised.

The present paper aims to address the question of grid-generation for oceanic/atmospheric modelling from a pragmatic, rather than theory-laden perspective. As such, it was not felt that a reproduction of existing proofs would necessarily aid the reader in this regard.

I propose to amend Section 2.5 (around line 3, page 8) to more explicitly refer interested readers to the companion paper (Engwirda and Ivers, 2016), and offer a more formal summary of theoretical results/proofs.

*5. The abstract is misleading. The code may be recently developed but the techniques used in it are not recent.*

I do not believe that the methods presented in the present work are preexisting.

[Figure]

The hybrid Frontal-Delaunay surface meshing techniques described here, able to guarantee worst-case bounds on element quality and sizing conformance are, in my view, new. I am not aware of another algorithm with the same properties — able to produce smoothly varying Voronoi-Delaunay grids with very high mean element quality (similar to advancing front type schemes), while also guaranteeing worst-case bounds on element angles and conformance (a'la standard Delaunay-refinement techniques).

Existing methods for unstructured oceanic/atmospheric modelling appear to either lack provable worst-case bounds [Jacobsen et al., 2013], or generally produce grids with somewhat lower overall quality [Lambrechts et al., 2008].

The combination of the Frontal-Delaunay scheme with a coupled hill-climbing optimisation strategy to generate 'well-centred' grids is also, in my view, new. A number of additional remarks regarding the generation and benefits associated with 'well-centred' grids are already included in response to the reviewers first comment above. As per my response to this earlier comment, I believe that by making these arguments more clearly and earlier in the revised submission, the impact and novelty of the approaches pursued in the present work will be more strongly articulated.

*6. There is not detail about the initialization on the sphere of the algorithm. You mention that the algorithm scans the triangles that do not verify given criteria, but how are these initial triangles created?*

Agreed. I propose amending Section 2.5 in the revised manuscript to contain the following information:

12 points describing a coarse regular icosahedron are initially projected onto the spheroidal surface at the beginning of the refinement process. Refinement then proceeds according to the Frontal-Delaunay scheme outlined in Section 2.5 (i.e. until all constraints — element shape, size — are satisfied).

Please let me know if you have further suggestions or comments regarding the sub-

mission or my responses to your review.

Kind regards,

Darren Engwirda

————————————————————

[Figure]

Geosci. Model Dev. Discuss.,
doi:10.5194/gmd-2016-296-AC3, 2017

[Figure]

Thank you for your comments and recommendations regarding the draft manuscript. I propose to incorporate the majority of your suggested changes 'as-is' in the revised submission. In all cases I have tried to respond in detail to your comments, including one or two instances where I have attempted to clarify the intent of the current draft.

I have found all of your suggestions to be helpful, and look forward to amending the manuscript in response to your review.

Overall, I agree with your suggestion that additional algorithmic detail should be pro-

vided, and will include such material in the revised submission. I intend to include additional details on the Frontal-Delaunay algorithm, in addition to brief pseudo-code descriptions of the full grid-generation process.

To better emphasise the novelty of the proposed approach, I additionally suggest a slight reorganisation of the paper. I suggest to move the discussion currently contained in Sections 2.6 (Figure 3 — staggered unstructured C-grid formulations) and pages 16–17 (benefits of 'well-centred' staggered orthogonal grids) to the beginning of Section 2. Such a change will better motivate the remaining discussions — explaining that such numerical schemes (i.e. the MPAS framework) require grids that are locally-orthogonal, centroidal and well-centred — a set of constraints that are, in the general case, difficult to satisfy and are not reliably achieved by conventional grid-generation techniques.

Reviewer comments are presented in italics, my responses are included in plain-text.

*There is a sufficiently detailed description of the techniques used to improve the mesh quality, but the frontal-Delaunay technique is just mentioned. It would be helpful if it is described in some detail, for the intention of this paper is to help the potential user to learn about the technical details of the algorithm, and it is still a bit difficult to do. It would be very helpful to present a schematic of the algorithm at the very beginning, followed by the description of separate steps. This is partly done for the iterative mesh quality improvement, but I think that the entire algorithm has to be included. Also I would advice to be more clear with what is new in the algorithm.*

*Fig 1. The coloring used is non-informative. I was struggling to see some familiar topographic features but I could not. I would recommend to omit the topographic height, it does not have any sense here, only distracts you reader. Same concerns other mesh figures.*

Agreed. I suggest to adopt a simple flat yellow colour-scheme for the ocean grid-cells, with the existing white-fill adopted for the land-cells. This combination appears to offer good visual contrast.

*Page 2, line 6 'A majority of ...' — I do not think this statement is correct. Many of practically used ocean circulation models use so-called tripolar meshes with Mercator-type stretching. Cubed sphere is used less frequently (the internal Rossby radius of deformation in the ocean is decreasing toward high latitudes, and meshes refined in high latitudes are more natural)*

I agree that both cubed-sphere and tri-polar type grids are currently in wide use. I will update this sentence accordingly.

*Line 10 'leading to significant...' If lon-lat mesh is used for ocean modeling, it is of course rotated, so that poles are on the land and there is no singularity. What is discussed has relevance to the atmosphere, but not to the ocean. Since your mesh examples are related to the ocean, the discussion creates misunderstanding.*

I agree that this is true for Earth-centric ocean-modelling based on current continental configurations. Our requirements are somewhat more expansive than this though, also seeking to encompass Paleo-Earth studies, aqua-planets, exo-planets, and etc. In such cases, the polar singularities of lat-lon grids lead to numerical issues as per the current discussion.

I will amend this paragraph to better explain context and requirements here.

*Page 3, line 5 The discussion here misses the point that FESOM, FVCOM, SLIM, Fluidity may work on general triangular meshes. SUNTANS (and its numerous pre-decessors) need orthogonal (well-centered) meshes (with the circumcenters inside respective triangles).*

I agree. I will update the text accordingly to note that such models offer more flexibility in terms of grid requirements.

*Line 29 I think the main point of Lambrecht et al. is that a care is taken of the shape of coastlines, resolution of passages etc. This question is not reflected in the manuscript (except for conclusions), although it presents the main challenge. It is quality of trian-*

[Figure]

*gles close to coastlines that is problematic in many practical cases.*

*I would also recommend to mention Admesh (DOI 10.1007/s10236-012-0574-0) which relies on the Persson's approach.*

Agreed. I will add a reference and brief description of Admesh and amend that of Lambretch et al.

The focus of the current work though is on multi-resolution global simulations, in which coastal constraints are typically neglected. The main challenge here is the generation of locally-orthogonal, centroidal and well-centred grids appropriate for unstructured C-grid schemes such as MPAS.

I look forward to incorporating coastal constraints in a future version of the algorithm.

*Page 4. Section 2. I would start here with brief description of the entire algorithm. Your reader keeps wondering what is the algorithm before the end of section 3. Present details of the Frontal -Delaunay algorithm and explain that in reality it is a 3D procedure with the central point of the sphere used to form the tetrahedra, and the restriction is just surface triangulation. Otherwise your preliminaries and Fig.2 are a bit embarrassing for a general reader (who would keep asking about $v_e$ and its relation to the surface).*

*Page 5 Line 13 Restricted Delaunay tessellation — try to explain this better, by using illustrations. I do not see your Fig. 2 to be of much help.*

*Page 6. It is not clear how mesh spacing functions are used in the mesh construction. Are they taken in account in the frontal procedure? It is not mentioned. Help you reader to clearly see the steps of the algorithm.*

I agree. I suggest updating Figure 2 — adding additional detail illustrating the underlying tetrahedral grid emanating from the sphere centre. I will also update the description of the 'restricted' Delaunay structure accordingly.

I will expand the description of the Frontal-Delaunay algorithm as suggested, explicitly

describing the 'off-centre' point-placement scheme and the way in which the mesh-spacing function is utilised. This material is already available in Engwirda and Ivers (2016), but I will include a summary of the algorithmic detail here.

A full and general description of the Frontal-Delaunay algorithm and it's underlying theory is provided in Engwirda and Ivers (2016), and I am conscious not to dwell on excessive detail here, but to instead focus on a practical application of the techniques to general-circulation modelling. I fully agree though that the reader may benefit from a little more detail, and I will update the manuscript accordingly.

*Page 7. Line 10 .. fully inverted elements? Please define what do you mean.*

In some cases, unconstrained updates to vertex coordinates and/or grid topology could 'invert' triangles, reversing their orientation. Such cases would represent a local 'tangling' of the grid, which is obviously invalid. I will include the explanation "reversing its orientation".

*Fig. 3. Is not it generally known?*

Various unstructured general circulation models stagger variables according to a variety of different strategies. The staggered unstructured C-grid scheme described here has proven to be particularly successful (as per the MPAS framework) and I feel it's description is useful to the reader. This particular arrangement of variables also helps explain the necessity of the constraints on the grids themselves — that grids are required to be locally-orthogonal, centroidal and well-centred.

As per my initial remarks, I propose to move this discussion to the beginning of Section 2, better contextualising and motivating the subsequent description of the grid-generation algorithm.

*Page 8 Line 16 Fluid velocity and vorticity are commonly at different locations.*

Agreed. I will change this line to: "...and the fluid velocity field and vorticity distribution represented at other spatially distinct grid-points."

[Figure]

*Page 9, Line 4 ... Mention that you mean C-grid type techniques. There are other possibilities.*

Agreed. I will change this line to: "In this study, the development of locally-orthogonal grids appropriate for staggered unstructured C-grid schemes..."

*Page9, line 7 'described previously'? It was not really described here.*

I agree. As per the responses above, I will included additional description of the Frontal-Delaunay algorithm.

*Beginning from line 9, there is discussion that is either well known or irrelevant to the mesh generation.Why do you need it?*

I believe a basic description of the generation and usefulness of Voronoi grids — especially the well-centred, centroidal variety — is of use to the general reader. There is currently not consensus as to the optimal staggering of variables for unstructured general circulation models — MPAS uses the locally-orthogonal C-grid scheme described here, FESOM2 uses a type of B-grid finite-volume approach defined using a (non-orthogonal) barycentric dual-cell. A variety of other approaches also exist, including finite-element type formulations.

Additionally, while the use and construction of locally-orthogonal staggered unstructured grids (i.e. Voronoi diagrams) is a well-known concept in computational geometry, it may be less so to general readers in the atmospheric/ocean-modelling communities.

As per my initial remarks, I propose to move this discussion to the beginning of Section 2, better contextualising and motivating the subsequent description of the grid-generation algorithm.

*Page 11. Please define all quantities in (7) and better explain how computations are implemented.*

I do not fully understand this comment. All quantities in equation 7 appear to be de-

fined, and the subsequent paragraphs describe the way that the local 'gradient-ascent' type optimisation step is implemented. Please let me know which variables/steps need clarification.

*Page 13, line 14 What is the grid-quality vector?*

The 'grid-quality vector' is an array of element quality 'scores', one for each triangle in the grid. An 'area-length' quality metric is used in this study. The paragraph at the beginning of Section 3 (page 9, line 30) contains a description of the grid-quality vector.

*Page 13, line 26 What is the lexicographical comparison?*

A lexicographical comparison is a 'worst-first' comparison of grid-quality scores. This term is defined on page 10, line 32, where it is first introduced in the manuscript. Use of this terminology is consistent with it use in Klinger and Shewchuk (2008).

*Page 20, Section 5. Please clearly define the novelty. What is describe is the selection of known steps.*

*Page 22 , line 14 (ii)–? I think it is secondary and technical issue. Well-centeredness and coastlines are real algorithmic questions.*

Unstructured general circulation frameworks, such as MPAS, require grids that are locally-orthogonal, centroidal and well-centred. It is not my experience that conventional grid-generation algorithms are successful in satisfying this set of constraints in the general case.

The JIGSAW-GEO algorithm introduced here has been shown to effectively produce such grids, allowing the multi-resolution capabilities of a framework such as MPAS to be better utilised. The use of complex grids with highly non-uniform spacing constraints, such as those shown in Figures 9–10, can now be investigated. Fully solution-adaptive simulations, as mentioned at page 22, line 17, can now also be explored as a result. I look forward to reporting on these studies in the future.

[Figure]

As per my initial comments, I suggest slightly rearranging the order of the paper — moving the discussion of 'local-orhogonality', 'centroidalness' and 'well-centredness' to the beginning of Section 2 and better emphasising the importance and difficulty of satisfying these characteristics in the general case. This change is intended to better contextualise the grid-generation techniques described here — showcasing their novelty and usefulness.

Please let me know if you have further suggestions or comments regarding the submission or my responses to your review.

Kind regards,

Darren Engwirda
* * *
[Figure]

Geosci. Model Dev. Discuss.,
doi:10.5194/gmd-2016-296-AC4, 2017

[Figure]

Thank you for your additional comments. I include my response below. Reviewer comments are presented in italics, my responses are included in plain-text.

*My main concern is still about the novelty of the method. You say:*

*In the current work, it's shown that a combination of Frontal-Delaunay refinement and hill-climbing optimisation is an effective strategy — able to produce very high-quality well-centred Voronoi-Delaunay grids even when complex, highly non-uniform grid sizing constraints are imposed. I believe this to be a new result of benefit to the unstruc-*

[Figure]

*tured oceanic/atmospheric modelling communities. Public availability of the associated JIGSAW-GEO grid-generator is also thought to be a further benefit to the community.*

*I do not believe that the methods presented in the present work are preexisting. The hybrid Frontal-Delaunay surface meshing technique described here, able to guarantee worst-case bounds on element quality and sizing conformance are, in my view, new. I am not aware of another algorithm with the same properties — able to produce smoothly varying Voronoi-Delaunay grids with very high mean element quality (similar to advancing front type schemes), while also guaranteeing worst-case bounds on element angles and conformance (a'la standard Delaunay-refinement techniques). Existing methods for unstructured oceanic/atmospheric modelling appear to either lack provable worst-case bounds [Jacobsen et al., 2013], or generally produce grids with somewhat lower overall quality [Lambrechts et al., 2008]. The combination of the Frontal-Delaunay scheme with a coupled hill-climbing optimisation strategy to generate 'well-centred' grids is also, in my view, new.*

*EVERY mesh generator (edge, face, volume) has a main engine (Delaunay, Frontal, Octree, coupled) and an optimization phase that follows [1], so there is nothing new to that. The facts that you apply it to oceanic/atmospheric communities or that it is publicly available do not make these techniques new. I have added a list of references on high quality surface mesh generation that present the same high quality based on the same techniques [2,3,4,5] on top of the coupled Delaunay advancing front variants which are not fully referenced. Feel free to include them or not.*

*Nevertheless, I completely agree with the fact that the application of these techniques to the oceanic/atmospheric communities is new and interesting and therefore, the paper should be published.*

*[1] Frey,P.J. and George,P.L., Meshing, applications to finite elements, Hermes, Paris, 1999.*

*[2] J. Tristano, S. Owen, S. Canann, Advancing front surface mesh generation in para-*

[Figure]

*metric space using a Riemannian surface definition, in: IMR, 1998, pp. 429–445.*

*[3] D. Rypl, P. Krysl, Triangulation of 3D surfaces, Eng. Comput. 13 (1997) 87–98.*

*[4] C. Lee, Automatic metric advancing front triangulation over curved surfaces, Eng. Comput. 17 (1) (2000) 48–74. 642–667.*

*[5] Löhner R. Regridding surface triangulations. Journal of Computational Physics 1996; 126:1–10.*

I agree with much of what is stated here. After also considering the comments of other reviews/readers, I suggest making a number of changes to address these concerns:

1. I am very happy to add reference to [1–5] as suggested. A brief discussion of these methods will be added in Section 2, throughout the description of the Frontal-Delaunay algorithm.

2. I will include a more detailed description of the Frontal-Delaunay algorithm, and additional pseudo-code descriptions for the full grid-generation process. I will explicitly describe the 'off-centre' point-placement strategy and the way in which it leverages the mesh-spacing function. This material is already available in Engwirda and Ivers (2016), but I will include a summary of the algorithmic detail here.

While the suggested references [1–5] all describe high-quality approaches for surface grid-generation, they do, in some cases, differ in the details. I aim to better position and contrast the methods described in the current work through this extended description.

3. To better emphasise the novelty of the proposed approach, I additionally suggest a slight reorganisation of the paper. I suggest to move the discussion currently contained in Sections 2.6 (i.e. Figure 3 and accompanying text — description of the staggered unstructured C-grid formulation) and pages 16–17 (benefits of 'well-centred' staggered orthogonal grids) to the beginning of Section 2.

This change will better motivate the remaining discussions — explaining that such

numerical schemes (i.e. the MPAS framework) require grids that are locally-orthogonal, centroidal and well-centred — a set of constraints that are, in the general case, difficult to satisfy and are not reliably achieved by conventional grid-generation techniques.

This change will therefore better showcase the performance of the algorithms presented in the current work — demonstrating that such locally-orthongonal, centroidal and well-centred grids can be generated for complex cases, such as the highly nonuniform grids shown in Figures 9–11. Such capability will allow the multi-resolution capabilities of a framework such as MPAS to be better utilised.

Please let me know if you have further suggestions or comments regarding the submission.

Kind regards,

Darren Engwirda
* * *

---

## Author Response (AR2)

To: The Topical Editor Geoscientific Model Development

Dear Editor,

Please find enclosed a revision of the manuscript: 'JIGSAW-GEO (1.0): Locally-orthogonal staggered unstructured grid-generation for general circulation modelling on the sphere' to be considered for publication in *Geoscientific Model Development*.

In addition to the initial revision of the paper (as detailed below), I have compiled a Zenodo archive of the JIGSAW-GEO library and updated the references accordingly.

Sincerely,

Darren Engwirda

Postdoctoral Associate Dept. of Earth, Atmospheric and Planetary Science Massachusetts Institute of Technology engwirda [at] mit [dot] edu

**JIGSAW-GEO** (1.0): Locally-orthogonal staggered unstructured grid-generation for general circulation modelling on the sphere\**

**Darren Engwirda1,2**

1Department of Earth, Atmospheric and Planetary Sciences, Room 54-1517, Massachusetts Institute of Technology, 77 Massachusetts Avenue, Cambridge, MA 02139-4307 2NASA Goddard Institute for Space Studies, 2880 Broadway, New York, NY 10025 USA *Correspondence to:* Darren Engwirda (engwirda@mit.edu)

**Abstract.** An algorithm for the generation of non-uniform, locally-orthogonal staggered unstructured spheroidal grids is described. This technique is designed to generate very high-quality staggered Voronoi/Delaunay dual-meshes appropriate for general circulation modelling on the sphere, including applications to atmospheric simulation, ocean-modelling and numerical weather prediction. Using a recently developed Frontal-Delaunay refinement technique, a method for the construction

- 5 of high-quality unstructured spheroidal Delaunay triangulations is introduced. A locally-orthogonal polygonal grid, derived from the associated Voronoi diagram, is computed as the staggered dual. It is shown that use of the Delaunay-refinement Frontal-Delaunay refinement technique allows for the generation of unstructured grids that satisfy a priori constraints on minimum mesh-quality. The initial staggered Voronoi/Delaunay tessellation is iteratively improved through very high-quality unstructured triangulations, satisfying a-priori bounds on element size and shape. Grid-quality is further improved through the
- 10 application of hill-climbing type optimisation techniques. Such an approach Overall, the algorithm is shown to produce grids with very high element quality and smooth grading characteristics, while imposing relatively low computational expense. Initial results are presented for a A selection of uniform and non-uniform spheroidal grids appropriate for high-resolution, multiscale general circulation modelling are presented. These grids are shown to satisfy the geometric constraints associated with contemporary unstructured C-grid type finite-volume models, including the Model for Prediction Across Scales (MPAS-O).
- 15 The use of user-defined mesh-spacing functions to generate smoothly graded, non-uniform grids for multi-resolution type studies is discussed in detail.

**Keywords.** Grid-generation; Frontal-Delaunay refinement; Voronoi tessellation; Grid-optimisation; Geophysical fluid dynamics; Ocean modelling; Atmospheric modelling; Numerical weather predication; Model for Prediction Across Scales (MPAS)

**1 Introduction**

20 The development of atmospheric and oceanic general circulation models based on *unstructured* numerical discretisation schemes is an emerging area of research. This trend necessitates the development of unstructured grid-generation algorithms for the production of designed to produce very high-resolution, *guaranteed-quality* unstructured triangular and polygonal

\*A short version of this paper appears in the proceedings of the 24th International Meshing Roundtable (?).